# Long-term trends in the ionospheric response to solar EUV variations

Rajesh Vaishnav[1], Christoph Jacobi[1], and Jens Berdermann[2]

[1]Leipzig Institute for Meteorology, Universität Leipzig, Stephanstr. 3, 04103 Leipzig, Germany
[2]German Aerospace Center, Kalkhorstweg 53, 17235 Neustrelitz, Germany

*Correspondence to*: Rajesh Vaishnav (rajesh_ishwardas.vaishnav@uni-leipzig.de)

**Abstract.** The thermosphere-ionosphere system shows high complexity due to its interaction with the continuously varying solar radiation flux. We investigate the temporal and spatial response of the ionosphere to solar activity using 18 years (1999-2017) of total electron content (TEC) maps provided by the international global navigation satellite systems service and twelve solar proxies (F10.7, F1.8, F3.2, F8, F15, F30, He II, Mg II index, Ly-$\alpha$, Ca II K, daily sunspot area (SSA) and sunspot number (SSN)). Cross-wavelet and Lomb Scargle periodogram (LSP) analysis are used to evaluate the different solar proxies with respect to their impact on the global mean TEC (GTEC), which is important for improved ionosphere modeling and forecasts. A 16 to 32-day periodicity in all the solar proxies and GTEC has been identified. The maximum correlation at this time scale is observed between the He II, Mg II, and F30 indices and GTEC, with an effective time delay of about one day. The LSP analysis shows that the most dominant period is 27 days, which is owing to the mean solar rotation, followed by a 45-day periodicity. In addition, a semi-annual and an annual variation were observed in GTEC, with the strongest correlation near the equator region where a time delay of about 1-2 days exists. The wavelet variance estimation method is used to find the variance of GTEC and F10.7 during the maxima of the solar cycles SC 23 and SC 24. Wavelet variance estimation suggests that the GTEC variance is highest for the seasonal timescale (32 to 64-day period) followed by the 16 to 32-day period, similar to the F10.7 index. The variance during SC 23 is larger than during SC 24. The most suitable proxy to represent solar activity at the time scales of 16 to 32-day and 32 to 64-day is He II. The Mg II index, Ly-$\alpha$, and F30 may be placed second as these indices show the strongest correlation with GTEC, but there are some differences in correlation during solar maximum and minimum years, as the behavior of proxies is not always the same. The indices F1.8 and daily SSA are of limited use to represent the solar impact on GTEC. The Empirical Orthogonal Function (EOF) analysis of the TEC data shows that the first EOF component captures more than 86% of the variance, and the first three EOF components explain 99% of the total variance. EOF analysis suggests that the first component is associated with the solar flux and the third EOF component captures the geomagnetic activity as well as the remaining part of EOF1. The EOF2 captures 11% of the total variability and demonstrates the hemispheric asymmetry.

## 1 Introduction

The interaction of solar radiation with the ionosphere is complicated due to several mechanisms with the potential to modulate the thermosphere-ionosphere (T-I) system at different time scales ranging from the 11-year solar cycle down to minutes (e.g., Liu et al., 2003; Afraimovich et al., 2008; Liu and Chen 2009; Chen et al., 2012). The ionosphere plasma response to solar EUV and UV variations has been widely studied using ground- and space-based observations (e.g. Jakowski et al., 1991; Jakowski et al., 1999; Jacobi et al., 2016; Schmölter et al., 2018), as well as by numerical and empirical modeling (e.g. Ren et al., 2018; Vaishnav et al., 2018a,b). These studies have shown that the response of the ionosphere to solar EUV radiation variations is delayed by 1-2 days at the 27 days solar rotation period (e.g. Jakowski et al., 1991; Jakowski et al., 2002; Afraimovich et al., 2008; Min et al., 2009; Lee et al., 2012; Jacobi et al., 2016).

To understand the underlying mechanisms of the delay observed in the ionospheric plasma, Jakowski et al. (1991) used a one-dimensional numerical model to explain the ionospheric delay of about 1-2 days. They concluded that the ionospheric delay could be attributed to the delayed atomic oxygen density variation at 180 km height produced via $O_2$ photodissociation. Ren et al. (2018) performed multiple numerical experiments using the TIE-GCM model to investigate the potential physical mechanisms responsible for the ionospheric delay. Their simulation results revealed that photochemical, dynamic, and electrodynamic processes, as well as the geomagnetic activity, can be associated with the ionosphere response time. Vaishnav et al. (2018a) performed CTIPe model simulations to explore the dominant mechanisms and suggested that transport might be the leading process responsible for the ionospheric delay.

Apart from solar radiation, the T-I system is also influenced by different external forces, which include lower atmosphere forcing, particle precipitation, geomagnetic, and solar wind conditions (e.g., Jakowski et al., 1999; Min et al., 2009). As a result, the ionospheric plasma behavior is continuously varying depending particularly on the solar activity conditions. Lean et al. (2016) constructed a statistical model and characterized the spatial patterns of the ionospheric behavior at different time scales arising from the solar and geomagnetic conditions and showing annual and seasonal oscillations. Medium-term and long-term ionospheric variability, ionospheric storm time response, as well as solar activity and geomagnetic response were discussed by Kutiev et al. (2013).

The mean solar rotation period is approximately 27 days, and therefore similar periodic variations are expected in the ionospheric parameters, such as total electron content (TEC, measured in TECU: 1 TEC Unit = $10^{16}$ electrons/m$^2$), NmF2, etc. (e.g., Min et al., 2009). Hocke (2008) studied oscillations in the global mean TEC (GTEC) and solar EUV (Mg II index) and reported dominant periodicity at the time scale of the solar rotation, the annual, semi-annual, and solar cycle. These oscillations observed in GTEC could be related to the ionizing radiation changes. Kutiev et al. (2012) studied the mid and low latitude ionospheric response to solar activity. They suggested that the 27 days periodicity as the main dominant oscillation during the study period.

In order to understand the variability of the T-I system, the knowledge of solar EUV variations is essential. Since direct EUV measurements before the space age were not available due to atmospheric absorption, solar

proxies are frequently used to represent solar variability till today. The most widely used proxies for ionospheric applications are the F10.7 index (solar radio flux at 10.7 cm, measured in solar flux units (sfu), see Tapping, 1987; Maruyama, 2010), the Mg II index (the core-to-wing ratio of the Magnesium K line; Maruyama, 2010), and indices based on direct EUV measurements (e.g., Schmidtke, 1976; Unglaub et al., 2011; Jacobi et al., 2016) such as the Solar EUV Experiment (SEE, Woods et al., 2000) onboard the Thermosphere Ionosphere Mesosphere Energetics and Dynamics (TIMED) satellite. Using the latter poses the potential problem of satellite degradation (BenMoussa et al., 2013, Schmidtke et al., 2015), which may be overcome by repeated calibration or in-flight calibration as had been applied during the SolACES experiment on board the ISS (Schmidtke et al., 2014, 2015). The understanding and realistic estimation of solar irradiance have been an open issue for long times, but in recent times the EUV datasets (either direct measurements, composite datasets or models) have considerably improved (e.g., Haberreiter et al., 2017).

This paper investigates and evaluates the correlation between GTEC and different solar EUV proxies in the time period January 1999 till December 2017. The purpose of utilizing several proxies is to estimate the respective correlation and the ionospheric delay to identify proxies which are most suitable for describing the solar-ionosphere relationship at different time scales and under different solar activity conditions. Therefore, the ionospheric delay at the different oscillation periods of solar irradiance is addressed to investigate GTEC response to solar variations as indicated by various solar proxies. To understand the variability in the ionosphere, we use the method of Empirical Orthogonal Functions (EOFs) in order to classify the temporal and spatial variability in the ionosphere.

## 2 Data Sets

Global TEC maps for the period 1999 to 2017 are available from the International Global Navigation Satellite Systems (GNSS) Service (IGS, Hernandez-Pajares et al., 2009). We used NASA's 2-hourly global TEC maps, which are available in IONEX format from the CDDIS (ftp://cddis.gsfc.nasa.gov/gnss/products/ionex/; Noll, 2010) data archive service (CDDIS, 2018). These maps are available in a spatial resolution of $2.5°$ in latitude and $5°$ in longitude. We selected 12 solar proxies for GTEC correlation analysis, namely the F10.7 index, the Bremen composite Mg II index (IUP, 2017), the Ca II K index, the daily sunspot area (SSA), the He II (Dudok de Wit, 2011), and F1.8, F3.2, F8, F15, F30 solar radio flux emission at 5 wavelengths (Dudok de Wit et al., 2014, Haberreiter et al., 2017) as well as Ly-$\alpha$ and SSN (sunspot number, Wolf, 1856) indices which are available from NASA's Goddard Space Flight Center through the OMNIWeb Plus database (http://omniweb.gsfc. nasa.gov/; King and Papitashvili, 2005). The F10.7 index data were taken from the LISIRD (DeWolfe et al., 2010) database, whereas F1.8, F3.2, F8, F15, F30, Ca II K index and daily SSA proxies are available from the SOLID database (http://projects.pmodwrc.ch/solid/; Schöll et al., 2016; Haberreiter et al., 2017). SOLID data were only available for the time interval 1999-2012 and all other data cover the full period from 1999 till 2017. The daily TEC and GTEC values were calculated from the gridded 2-hourly TEC maps to obtain a time resolution corresponding to those of the solar proxies. Further, to investigate the relation between GTEC and geomagnetic activity, we have used daily Kp, Dst, and Ap indices, which were taken from the OMNIWeb Plus database. To calculate the cross-correlation between solar proxies and GTEC,

we used the wavelet cross-correlation analysis, cross-correlation sequence, and Pearson cross-correlation method.

## 3 Results and discussion

### 3.1 Long-term variations of TEC and EUV flux

In the following, we analyze the long-term variations of GTEC and EUV flux for the period 1999 till 2017,
which partially covers the solar cycles (SC) 23 and 24. The temporal variation of the zonal mean TEC is shown in Fig. 1(a). In SC 23, the TEC values at low latitudes reach up to 80 TECU, while during SC 24 TEC was considerably smaller, which confirms that the zonal mean TEC behavior is strongly depending on the solar activity, as the solar activity was very low during SC 24 compared to SC 23. The amount of free electrons in the ionosphere mainly depends on the photoionization of atomic and molecular neutrals due to solar EUV radiation
along with the recombination at different height and solar zenith angle. The lowest TEC values are observed in the years 2008 and 2009 during the extended solar minimum of SC 23 (Nikutowski et al., 2011). From the zonal mean plot (Figure 1(a)), temporal variations are visible, which result from annual and semi-annual variations in the ionosphere. Figure 1(b) shows the normalized time series of GTEC and twelve solar proxies for the available data in the analyzed time period 1999-2017. Note that Emmert et al. (2017) showed that GTEC values before
2001 are lower than values observed later. This effect should, however, be of minor importance for our analyses below. As the ionosphere response to solar radiation varies for different wavelengths, we used twelve solar proxies based on different measurement techniques and spectral characteristics. Hocke (2008) analyzed GTEC and Mg II index observations and showed that 1% change in Mg II index results in about 22% change in GTEC. As all the time series in Fig. 1 show a similar overall variation during the 11-year solar cycle. The fundamental
behavior of solar radiation emission is not identical at all wavelengths and thus for all solar proxies, as the plasma heating and atomic processes are different (Dudok de Wit et al., 2014) but the long-term trends and variations look similar for all the proxies shown here.

Figure 2 shows the spatial variation of TEC averaged over the period 1999 to 2017, where the superimposed white contour lines show the standard deviation calculated from the daily TEC data. The magnetic equator is
indicated by a dashed black line. A similar analysis has been shown by Guo et al. (2015) using the same TEC dataset within the period 1999 till 2013, finding a comparable spatial distribution. The maximum TEC values are distributed along the equator around $\pm20^{\circ}$ and decrease towards the poles. Maximum values of the standard deviation are observed in the low latitude region with about 15 TECU. The spatial distribution of TEC depends on the ionization of neutrals, transport processes, and recombination, which varies with latitude and longitude.

Note that the T-I system is not only influenced by the solar electromagnetic radiation but also by changing solar energetic particles and geomagnetic conditions due to solar wind variations or Coronal Mass Ejections reaching the Earth (e.g., Abdu 2016; Tsurutani et al., 2009). The response to solar forcing is higher during solar maximum when the interaction of the solar wind with Earth´s upper atmosphere causes ionospheric disturbances

at high latitudes along magnetic field lines visible in enhanced TEC values. During solar maxima, the T-I regime can partially be controlled by the solar wind activity superseding the solar radiation impact. However, during periods of low solar activity, the local variability in the ionosphere is also not only regulated by the solar radiation but can be influenced by lower atmospheric forcing (e.g. Forbes et al., 2000; Koucká Knížová et al., 2015) and by the solar wind, in particular from coronal holes (e.g., Zurbuchen et al., 2012; Verkhoglyadova et al., 2013).

### 3.2 Spectra of GTEC and solar proxies

The datasets mentioned above are used to analyze the oscillatory behavior of the T-I system. The periodicities in the solar proxies have been studied by various authors to explore the response of the terrestrial atmosphere and especially the T-I region to solar variability. Here we will investigate and compare the different temporal patterns of GTEC and multiple solar proxies, since proxies may differ in their periodicity depending on the underlying source mechanism.

The cross-wavelet technique from Grinsted et al. (2004) was applied, where Morlet wavelets were used as mother functions. The cross-wavelet technique allows to indicate common high-power regions between two time series. This allows us to determine the dominant correlated oscillations of the ionosphere and important solar proxies. The cross-wavelet analyses of GTEC with four selected solar proxies (F10.7, Mg II, SSN, and Ly-$\alpha$) are shown in Fig. 3. The most dominant periods observed are in the 16 to 32-day interval visible in all GTEC solar proxy relations during solar maxima. This is, however, not the case during solar minimum when the solar driven ionospheric variation is lower due to lower solar activity, and the influence of other dynamical processes in the ionosphere (e.g. lower atmospheric forcing) is stronger. Another high-power region is visible in the 128 to 256-day period, representing the semi-annual oscillations in both GTEC and solar parameters. The semi-annual oscillation is mostly dominant during the solar maximum years 2001-2002 and 2011-2012. The black arrows in Fig. 3 indicate the phase relationship between solar proxies and GTEC, with in-phase (anti-phase) relation shown by arrows pointing to the right (left), while downward (upward) direction means that GTEC is leading (lagging). As expected, in the region of 16 to 32-day GTEC is broadly in phase with the solar proxies, whereas this behavior is not consistent at the semi-annual (128 to 256-day) and annual (256 to 512-day) period ranges. The most dominant joint annual oscillations are observed between GTEC and Ly-$\alpha$. The annual oscillation can be found mostly during solar maximum.

To examine the oscillatory behavior of GTEC and solar proxies more precisely, the Lomb Scargle periodogram (LSP, Lomb, 1976; Scargle, 1982) technique was used. The corresponding spectral analysis is shown in Fig. 4. Here, the power was normalized and converted into a logarithmic scale, and the 95% confidence level is added to each spectrum as a dashed blue line. The curves have been vertically offset by 15. In this analysis, data from 1999 to 2012/2017 was used. The dominant frequencies observed in GTEC are 27 days, annual, and semi-annual, which is in line with Hocke (2008). Clearly visible in all the solar proxies as well as in GTEC are the mean solar rotation period of about 27 days. Pancheva et al., (1991) showed that the 27-day variation in the

lower ionosphere (D-region) is often predominantly caused by dynamical forcing (planetary waves), not by direct solar forcing, particularly in winter under low solar activity. However, the D region ionization contributes only weakly to TEC. A 45-day periodicity is observed GTEC, F10.7, Mg II, and SSN. A 45 days periodicity was reported in various solar proxies (Lou et al., 2003; Kilcik et al., 2016, 2018; Chowdhury et al., 2015) using LSP and wavelet analysis. Lou et al. (2003) reported a period of about 42 days in X-Ray solar flares during SC 23. Kilcik et al. (2018) analyzed sunspot counts in flaring and non-flaring active regions for SC 23 and 24 and observed a 45 days periodicity in flaring active regions. They concluded that a 45 days period is one of the fundamental periods of flaring active regions. A similar periodicity was observed during SC 24 by Chowdhury et al. (2015) in SSA, SSN, and the F10.7 index.

In the Mg II index, which is widely used to represent the solar variability, the dominant periods observed are 27 days, and its 2nd harmonic 13.5 days also described by Hocke (2008). Here the same oscillation is also visible in the Ly-α spectrum. In the F1.8 index, the annual frequency is observed. A semi-annual oscillation is seen in GTEC. This variation is associated with a dynamical effect of the atmosphere (Liu et al., 2006). Note that the wavelet spectra show some periodicity at the half-year time scale for GTEC and F30, but with variable phase so that they extinguish in the periodogram. 128- and 256-day periodicities were reported by various authors (Lou et al., 2003; Kilcik et al., 2014, 2018; Chowdhury et al., 2009). Lou et al. (2003) reported a 259±24 days variation in M5 class X-ray flares during the solar maximum of SC 23. This periodicity may be attributed to non-flaring active regions and developed sunspot groups (Kilcik et al., 2018). Further, Kilcik et al. (2014, 2018) confirmed that the 128 days periodicity is one of the characteristic periodicities of solar flares and also flaring active regions.

### 3.3 Wavelet Cross-Correlation

To evaluate the relation between the solar proxies and GTEC, we analyzed the wavelet cross-correlation for the different periods 8 to 16-day, 16 to 32-day, 32 to 64-day, and 64 to 128-day using the wavelet cross-correlation sequence method based on the Maximal Overlap Discrete Wavelet Transform (MODWT) technique (Percival and Walden, 2000). The MODWT technique is a modified version of the discrete wavelet transform from Mallat (1989). In Fig. 5 these cross-correlation coefficients are indicated by the background color, while the inserted numbers show the ionospheric delay in days. The delay is mostly positive, which means that TEC is following the solar proxies. On the 8 to 16-day time scale, maximum correlation is found for He II with a correlation coefficient of about 0.62, and the second maximum correlation is observed for the F15 index, both with a lag of about one day. The lowest correlation of about 0.25 is found for the F1.8 index. Compared to the 8 to 16-day period range, the 16 to 32-day period shows a much stronger correlation with more than about 0.6 for all the proxies. Here a maximum correlation of about 0.9 is observed for the He II and Mg II index, with a GTEC delay of about one day. The F30 index and the Ly-α index also shows a strong correlation. The lowest correlation of 0.59 is seen for the daily SSA. A similar result can be observed in the 32 to 64-day period range. Here, maximum correlation is observed again for the He II and Mg II indices having a correlation coefficient of 0.9 and a delay of about two days. Another particular strong correlation of about 0.8 is observed with Ly-α and

Ca II K having a GTEC delay of about one and two days, respectively. Only a weak correlation of about 0.5 with small GTEC lag time is seen for the daily SSA. The similar behavior in the 16 to 32-day and 32 to 64-day intervals is owing to the fact that the 27-day periodicity is only a mean value of the solar differential rotation. It also strongly depends on the lifetime and proper motion of the observed active regions. This results in strong correlations observable also in the 32 to 64-day interval. In the 64 to 128-day interval, a longer time lag is reached with above five days for several proxies. Here the maximum correlation is found for the He II index with about 0.6 and weakest correlation is seen with about 0.4 for the F1.8 index. Generally, the Mg II and He II proxies show the strongest correlation with GTEC for all period intervals. A strong correlation is also seen for Ly-$\alpha$ and F30, while the weakest correlation is seen for F1.8 and the daily SSA. Figure A.1 in the appendix shows the correlation between solar proxies and GTEC at zero lag at different time scales. Like Fig. 5 it shows strong correlation for Ly-$\alpha$ and F30.

Figure 6 shows the wavelet variance estimated for GTEC and F10.7 using the MODWT technique with the Daubechies 2 (db2) wavelet filter. Here we have selected the time series from the years 2000 to 2002 (maximum of SC 23) and 2012 to 2014 (maximum of SC 24). The red/black color in the plot represents the SC 23/SC 24 maximum. In GTEC, maximum variance appears in the 64 to 128-day interval, which is about a quarterly annual oscillation and belongs to the seasonal cycle, during SC 23. The second strongest variance is observed at the 16 to 32-day interval. A generally stronger variance can be observed in SC 23 compared to SC 24 for all the analyzed period intervals. In the case of the F10.7 index, the maximum variance is visible at the 16 to 32-day interval, which here shows a predominant variance for the solar rotation period. As expected, no significant semi-annual cycle is visible. Here again, the observed variance during SC 23 is stronger compared to SC 24.

## 3.4 Influence of the solar activity on GTEC

This work aims to understand the interaction between solar radiation and the T-I system, especially at the time scale of the solar rotation. To scrutinize the consequence of different solar activity levels on the T-I system for short, and intra-annual (included all variability) time scales, we evaluate the running cross-correlation analysis between GTEC and solar proxies as shown in Fig. 7. The upper panel of Fig. 7 shows the running correlation for the short time scale. To calculate the short-term variation at the solar rotation period, the 27 days residual has been calculated by subtracting the 27 days running average values from corresponding datasets of GTEC and solar proxies (Mg II, SSN, and Ly-$\alpha$). The running correlation is calculated using the filtered time series at the solar rotation period by using a 365-day running window. The 365-day running mean Mg II index is added to show the overall solar activity in the upper and lower panel of Fig. 7. The correlation is likely to vary with respect to solar activity. Lower correlation is observed during low solar activity. A similar kind of analysis was shown by Chen et al. (2012).

Furthermore, to understand the relation between GTEC and solar proxies at longer timescales, we calculate the cross-correlation between the annual means. The maximum correlation is observed is about 0.93 (Fig. not shown) between the solar proxies and GTEC. Hence in comparison to short time scales variations, solar proxies are strongly correlated with GTEC.

The lower panel of Fig. 7 shows correlation at the intra-annual timescale, which includes all the variations i.e. seasonal, daily, and solar rotation. Here, a 365-day running window is used to calculate the running correlations based on unfiltered data. The correlation with all the solar proxies is smallest during the extended low solar activity phase during the solar minimum in 2008-2009. All solar proxies show similar behavior during low activity conditions: while the temporal variation of the correlation coefficient for Mg II, and Ly-$\alpha$ is largely similar, the SSN (blue curve) shows significantly different behavior. The strongest correlation is observed during the rising part of solar cycle 24. In comparison to all the other solar proxies, Mg II and Ly-$\alpha$ show a stronger correlation with GTEC, while the lowest correlation is given for SSN at short and intra-annual time scales.

Solar EUV variations can be well described by the solar proxies (e.g., F10.7, SSN) at the 11-year solar cycle variations but it shows weak correlation at short time scales (daily, 27 days solar rotation period) (e.g., Floyd et al., 2005; Chen et al., 2012) as shown in Fig. 7. At longer time scales, solar EUV and solar proxies are mainly controlled by solar magnetic activity. But at short time scales, these parameters vary differently as they originate from different excitations mechanisms in the solar surface (e.g. Lean et al., 2001; Chen at al., 2012).

Figure 8 shows the cross-correlation analysis of (a) F10.7 and (b) Mg II with the global, northern hemisphere (NH), southern hemisphere (SH), low (LL, $\pm30°$), middle (ML, $\pm$ ($30°$-$60°$)) and high (HL, $\pm$ ($60°$-$90°$)) latitude mean TEC. Generally, the correlation coefficients and the lag for the global, NH, SH, LL, and ML are very close to each other. The maximum correlation is found for GTEC and LL TEC with correlation coefficients of about 0.7 (F10.7) and 0.82 (Mg II) for a time delay of about two and one days, respectively. Generally, GTEC variability is mainly determined by the LL electron content, so that it is expected that the correlation coefficients for GTEC and LL are similar. The weakest correlation is observed for HL with a maximum correlation coefficient of 0.42 (F10.7) and 0.53 (Mg II) and a corresponding ionosphere response time of about two and one days, respectively (marked with a red star). NH and SH are comparable with slightly smaller correlations for SH. There is a weaker correlation for ML compared to LL, but the difference is not as large as the one for HL. Running correlations at intra-annual timescales, similar to Fig. 7 are shown in Fig. A.6 in the appendix.

Figure 9 shows the cross-correlation analysis between GTEC and solar proxies separately for each year at the timescale of 16- to 32-day. To calculate the wavelet cross-correlation, the data is filtered for different time scales using the MODWT. The upper panel of Fig. 9 shows the 365 days running mean F10.7. The delay is given as numbers inserted on the color-coded cross-correlation for the different solar proxies and time periods. As in Fig. 7, the overall trend shows that the correlation is weak during solar minimum and strong during high solar activity periods. The time delay is ranging between zero to three days for all solar proxies, but without obvious regularity with respect to the proxies or the time. As in Fig. 5, a generally strong correlation is found for He II and Mg II, while daily SSA and F1.8 indices show the weakest correlation. During the years of low solar activity 2007-2010, an especially weak correlation is visible for F3.2, F1.8 and Ca II K. The maximum F10.7 index is observed during 2001 with about 181 sfu. During the high solar activity years from 1999 to 2003 and from 2012 to 2014 a strong correlation of about 0.85 is observed for Mg II, He II, F30 (1999-2003, 2012), and Ly-$\alpha$ except for 2001, when the maximum annual mean F10.7 index is observed. During the maximum of

SC 23, the cross-correlation between Mg II, He II, Ly-α, F3.2, F15, and GTEC is about 0.7 with a delay of about 1-2 days. A weak correlation is observed for F1.8 and Ca II K. During low solar activity (years 2006-2010, 2016) when the average F10.7 index is below 75 sfu, a stronger correlation is observed between He II and Mg II and GTEC, with a correlation coefficient of more than 0.6. Only a weak correlation during low solar activity is observed for F1.8, Ca II K, F3.2, and daily SSA. During moderate solar activity years (2004-2005, 2011, 2015), when the average F10.7 index is about 90-120 sfu, Mg II, He II, F30, and Ly-α shows stronger correlation with GTEC with a delay of about one day. In summary, during low solar activity, most of the solar proxies show a weak correlation with GTEC but strong correlation is found for high solar activity. In comparison to other solar proxies, F1.8 and SSA are weakly correlated with GTEC. Figure A.2 in the appendix shows the correlation at 16 to 32-day time scale between solar proxies and GTEC at zero lag. It shows a similar correlation as Fig. 9. In comparison to the 16 to 32-day time scales, we further analyzed the cross-correlation and delay at 32 to 64-day time scales (Fig. A.3 in the appendix).

In summary, the most suitable proxy to represent the solar activity at the time scale of 16 to 32-day and 32 to 64-day during low, moderate, and high solar activity is He II. The Mg II index, Ly-α, and F30 also show a strong correlation with GTEC, but there are some differences in correlation during solar maximum and minimum years, as the behavior of proxies is not always the same. The F1.8 and daily SSA cannot adequately represent the solar activity at the solar rotation (16 to 32-day) time scale. As discussed above, solar proxies are more weakly correlated at shorter time scales than at longer time scales.

**3.5 Spatial distribution of the ionospheric response time**

Here we investigate the inter-annual spatial variability of the ionospheric response to solar variations. Figure 10 shows correlation and time lag between TEC and Mg II globally for a TEC map resolution of 2.5° in latitude and 5° in longitude. The left column shows yearly zonal means, while the right column shows 1999 to 2017 means with longitudinal resolution. The contour maps in the upper panel (lower panel) show the cross-correlation (time delay) where the inserted contour lines represent the standard deviation. Maximum correlation of about 0.9 is observed during high solar activity years at low latitudes. Figure 10(a,b) shows that the correlation decreases from low to high latitudes. In the NH, the correlation is the weakest south of the auroral oval, probably due to the fact that particle precipitation also changes with solar wind dynamics. Figure 10(c) shows the zonal mean time delay for the year 1999 to 2017, which is about one day in the low- and mid-latitudes. The delay generally increases towards high latitudes with a few exceptions occurring during low solar activity. There is a tendency that during high solar activity, the delay is increased slightly at low latitudes, but strongly (up to 3 days) in the high latitude region. A negative delay is observed during low solar activity, presumably associated with the meteorological effects as suggested by Ren et al. (2018). Another possible reason is ionospheric saturation, which might reduce the transport process during high solar activity due to lower recombination rates. Transport is one of the most critical parameters that control the behavior of the ionosphere. These results suggest that interannual variability depends not only on the solar activity but also on several other physical processes such as geomagnetic activity (Rich et al., 2003) and local ionospheric parameters such as neutral wind and lower

atmospheric forcing through the vertical coupling. Lee et al. (2012) analyzed electron density measurements from CHAMP and GRACE along with Global Ionosphere Maps (GIM) TEC maps in relations to the F10.7 index and showed the spatial distribution of delay and correlation coefficient during the years 2003 to 2007. They found a strong (weak) correlation between GIM TEC and F10.7 at the mid (high) latitude region, with a time delay of about 1-2 (2-4) day(s) which confirms qualitatively our results. Figure 10(d) shows the spatial distribution of the time delay, where an overall time delay of about one day with a standard deviation of less than one day is visible. The time delay is longer for the high latitude region, whereas the cross-correlation is weaker as can be seen in Fig. 10(b). In this region, the standard deviation is more than one day.

### 3.6 EOF analysis of ionospheric TEC

Ionospheric TEC is varying diurnally, daily, seasonally, on a decadal scale, as well as in latitude and longitude. To examine the spatial variability of TEC, we applied the Principal Component (PC) Analysis for signal decomposition (Preisendorfer 1988, Bjornson and Venegas, 1997) using EOFs, which decompose data into orthogonal modes of variability caused by solar and geomagnetic activity. The method is used to decompose the spatial-temporal field of TEC (time, longitude, latitude) into EOF components. To this end, we first calculate the data covariance matrix by using the TEC datasets, followed by finding the eigenvalues and corresponding eigenvectors (the EOFs). The explained variance of the k-th EOF is the corresponding eigenvalue divided by the sum of all eigenvalues. The PC is found by projecting the TEC anomalies onto the EOF. This method has been used to represent the variability in the T-I system and for T-I modeling (e.g., Zhao et al., 2005, Matsuo et al., 2012, Ercha et al., 2012, Anderson and Hawkins, 2015, Talaat and Zhu, 2016).

We analyzed the TEC data sets in a spatial grid of 71×72 (latitude and longitude) and a temporal length of 6940 days. Figure 11 shows the first four EOF maps in the upper panel followed by the PCs (middle panel) and the corresponding wavelet spectra (lower panel). The first three EOFs are similar to those presented by Talaat and Zhu (2016). The first EOF component explains approximately 86% of the variance. A high variability in the low latitude region and a smaller one at higher latitudes is shown. EOF1 shows the spatial distribution of TEC variance in general and is positive everywhere. This indicates a joint in phase variability of the entire ionosphere, which is larger at low latitudes. Consequently, as is shown in the middle panel of Fig. 11, its temporal amplitude varies from positive to negative following the solar activity and the annual and semiannual cycle. In the lower panel of Fig. 11, the wavelet analyses for the EOFs are shown. To get clear periodicity from the wavelet, we have used log2 of the power. Negative (positive) values indicate low (high) power. The wavelet analysis of EOF1 shows a 27 days periodicity associated with the solar rotation period. Annual and semi-annual oscillations are observed especially during the high solar activity years. The EOF2 captures 11% of the total variability and demonstrates a hemispheric asymmetry. The corresponding PC and wavelet analysis show a strong annual variability connected with seasonal variability and larger TEC during winter.

EOF3 captures about 1.79% of the total variability. EOF3 might be associated with non-solar effects and fine structures of the solar activity response, which is not captured by the first EOF as suggested by Talaat and Zhu

(2016). Note that EOF3 essentially shows a semi-annual and a relatively strong 27-day variability. EOF4 contributes with only 0.4% of the total estimated variability. Its shape is strongly non-zonal and reflects variations in longitudinal differences of the equatorial ionization anomaly. In the wavelet analysis, weak semi-annual and annual oscillations are visible. Note that the PC4 displays a possible long-term trend, which may indicate an effect of the secular change of the main magnetic field of the Earth. The oscillating structure of the EOF4 over the Atlantic resembles the results from numerical simulations by Cnossen et al. (2013).

In summary, the first two components capture almost 98% of the TEC variance, while the third and fourth component only contributes about 2%. This is similar to results of Zhao et al. (2005), Anderson and Hawkins (2015), and Talaat and Zhu (2016) who reported that more than 95% of the variance is explained by the first three EOFs.

In order to check the relation between solar proxies and geomagnetic parameters (daily Kp, Dst, and Ap indices) with PCs corresponding to EOFs, the wavelet cross-correlation and delay are shown in Fig. 12. In Fig. 12 the color indicates the maximum correlation coefficient, and the numerical values indicate the corresponding time delay in days. A strong correlation between PC1 and Mg II (F10.7) is observed with a coefficient of about 0.87 (0.8) and a time delay of one (two) day(s). This represents the strong correlation between global TEC and solar variability as PC1 is associated with solar variability. The geomagnetic parameters are generally more loosely connected with ionospheric variability, indicating the relatively fast ionospheric storm reaction compared to the longer lasting equatorial magnetic field depletion. PC3 shows a relatively strong correlation with the geomagnetic parameters, which indicates that this component, besides the remaining part of solar variability not included in EOF1, captures the geomagnetic activity effect on TEC. Here the Kp and Ap indices show a positive correlation of about 0.6 with a delay of about two days. In comparison to this, a negative correlation of about 0.7 is observed in the Dst.  Figure A.4 in the appendix shows the correlation between PCs and solar and geomagnetic proxies at zero lag. Furthermore, running correlations at interannual timescales, similar to Fig. 7 are shown in Fig. A.7 in the appendix using PCs and solar and geomagnetic parameters.

To assess the variability on the time scale of the solar rotation period, we filtered the GTEC time series in a period range of 25 to 35 days using a digital bandpass filter. The filtered time series is then used to compute EOFs. Figure 13 shows the first four EOF components in the upper row, and their corresponding wavelet transforms in the lower row. The first component captures almost 85.50 % of total variability, and it seems to be associated with solar activity. EOF1 shows high variability in the equatorial region. EOF2 captures 8.91 % of variability, and it is partly associated with hemispheric variability. EOF3 captures the variability of 4.92 %, which is not captured in the EOF2 component (in particular the hemispheric asymmetry). EOF2 does not show a clear hemispheric signal anymore, while EOF3 now does. The lower panel shows the corresponding wavelet spectra of PCs. Wavelet analysis shows clearly the expected periodicity in the 16 to 32-day period in all the PCs, with a response to the 11 years solar cycle.

Furthermore, wavelet cross-correlation analysis has been performed to understand the relation between solar proxies and geomagnetic parameters (with PCs corresponding to EOFs of the filtered data, as shown in Fig. 13) and shown in Fig. 14 (Fig. A.5 in the appendix for zero lag). It shows a similar kind of results as in Fig. 12 in the case of PC1. PC2 and PC3 are associated with geomagnetic activity. As compared to Fig. 12, PC2 shows strong correlation with magnetic indices. So, the distribution of variance is different here. This is because the coupled low-latitude/high-latitude magnetically forced variability, which is mainly represented by PC3 in the case of unfiltered data, is now distributed among PC2 and PC3 for the solar rotation period.

## 4. Summary

We have investigated the long-term ionospheric response during different solar activity, different timescales and spatial variations using twelve solar proxies (F10.7, F1.8, F3.2, F8, F15, F30, He II, Mg II index, Ly-$\alpha$, Ca II K, daily SSA, and SSN) and 18 years (1999-2017) of IGS TEC data. The cross-wavelet and LSP methods were used to examine the oscillatory behavior. The cross-wavelet analysis represents the 16 to 32-day period in all the solar proxies and GTEC. The maximum correlation with GTEC is observed between the He II index, Mg II index, and F30 in the period range of 16 to 32-day along with a time delay of about one day. Wavelet variance estimation suggests that GTEC variance is high for the 64 to 128-day interval followed by 16 to 32-day, while the F10.7 index is showing high variance for the 16 to 32-day interval.

Interannual variation of the cross-correlation analysis suggests that the correlation is varying with the solar activity. The most suitable proxy to represent the solar activity at the time scales of 16 to 32-day and 32 to 64-day during low, middle, and high solar activity is He II. The Mg II index, Ly-$\alpha$, and F30 may be placed at the second as these indices show a strong correlation with GTEC, but with some differences between solar maximum and minimum. The F1.8 and daily SSA poorly represent the solar activity effect on TEC. The spatial distribution of cross-correlation and time was estimated using the Mg II index. The results show significant temporal and spatial variations. Stronger correlation is observed near the equatorial region with a time delay of about 1-2 days. The magnetospheric inputs probably strongly influence both high and low latitude regions, but with a different sign.

TEC datasets also have been decomposed using EOFs along with the principal components analysis method to signify the spatial and temporal variation. The first EOF component captures more the 86% of the variability, and the first three EOF components explain 99% of the variance. EOF analysis suggests that the first component is associated with the solar flux and the third EOF component captures the geomagnetic activity as well as the remaining part of EOF1. The EOF2 captures 11% of the total variability and demonstrates the hemispheric asymmetry.

*Data availability.* IGS TEC data are provided via NASA through ftp://cddis.gsfc.nasa.gov/gnss/products/ionex/ (CDDIS, 2017). Daily F10.7 index can be downloaded from http://lasp.colorado.edu/lisird/noaa_radio_flux. Mg II index data are provided by IUP at http://www.iup.uni-bremen.de/UVSAT/Datasets/mgii (IUP, 2017). Solar proxies F30, F15, F8, F3.2, F1.8, Ca II K index, and daily SSA are available from the SOLID database (http://projects.pmodwrc.ch/solid/). The SSN, Ly-$\alpha$, Kp, Dst, and Ap indices are provided by NASA´s Goddard Space Flight Center through https://omniweb.gsfc.nasa.gov.

*Author contributions.* CJ, RV, and JB designed the study. RV analyzed the data. RV drafted the first version of the manuscript. All authors discussed the results and provided critical feedback and contributed to the final version of the manuscript.

*Competing interests.* C. Jacobi is one of the Editors-in-Chief of Annales Geophysicae. The authors declare that they have no conflict of interest.

*Acknowledgements.* We kindly acknowledge for the provision and NASA for providing of IGS TEC data (NASA), through ftp://cddis.gsfc.nasa.gov/gnss/products/ionex/ (CDDIS, 2017) and the daily F10.7 index (NOAA), Mg II data (IUP), solar proxies F30, F15, F8, F3.2, F1.8, Ca II K index, daily SSA (SOLID database) and SSN, Ly-$\alpha$, Kp, Dst, and Ap indices (NASA's Goddard Space Flight Center). The study has been supported by Deutsche Forschungsgemeinschaft (DFG) through grants No. JA 836/33-1 and BE 5789/2-1.

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

**Figures**

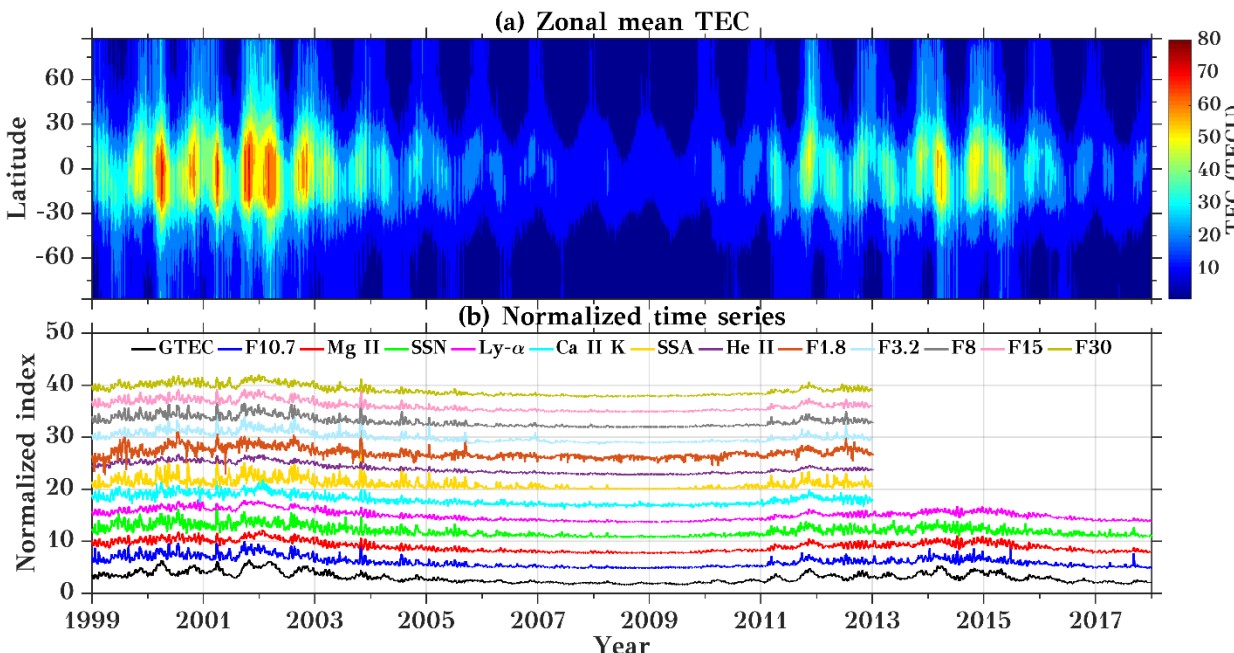

Figure 1: Time series of (a) zonal mean TEC and (b) smoothed normalized datasets of GTEC and different solar

630 proxies for the years 1999 to 2017. The curves in (b) are vertically offset by 3 each. X-axis labels refer to January 1st of the respective year.

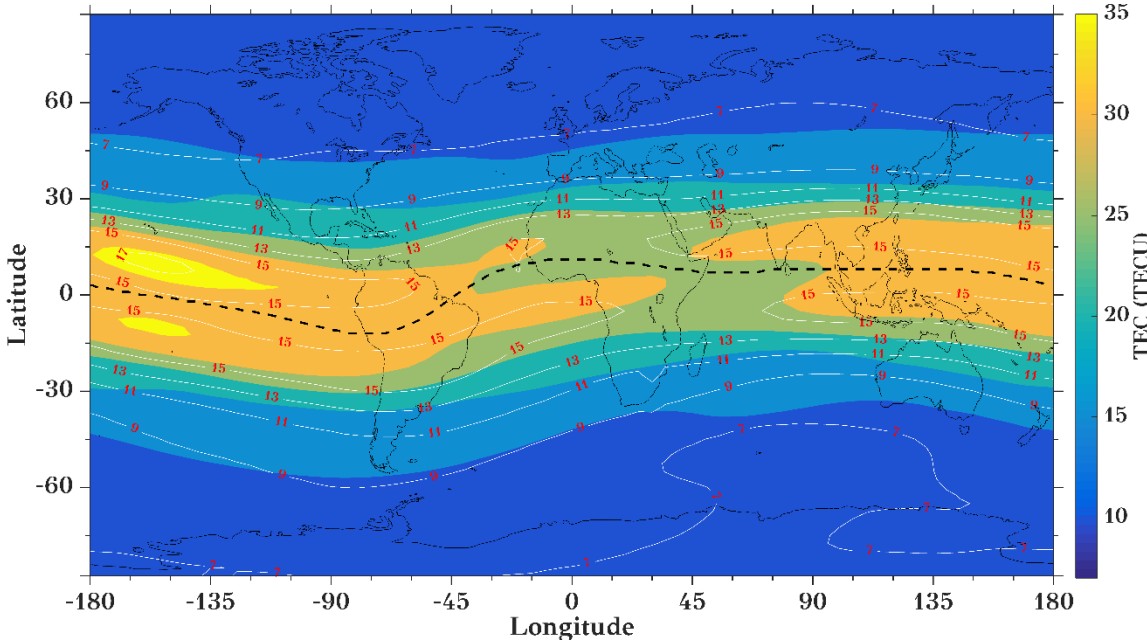

Figure 2: Long-term diurnal and annual mean TEC distribution during the years 1999 to 2017. The white contour lines indicate the standard deviation based on daily data. The black dashed line represents the magnetic equator.

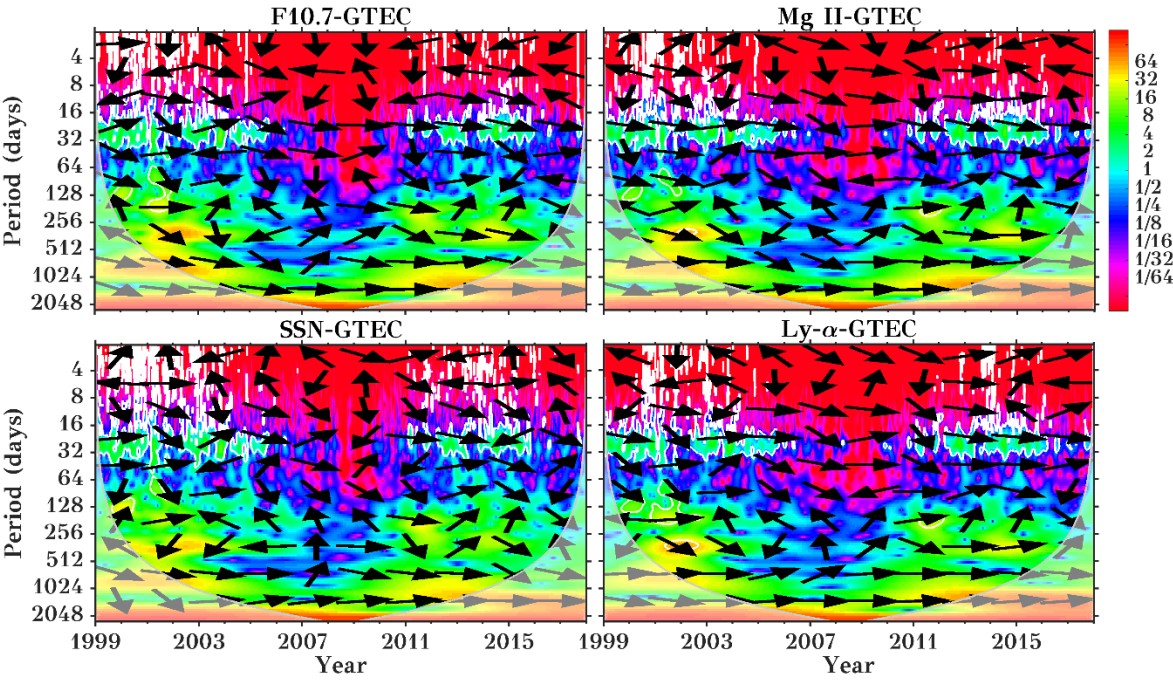

Figure 3: Cross-wavelet spectra for GTEC and different solar proxies during the years 1999 to 2017. The thin gray line shows the cone of influence, where a white line surrounds significant values. The arrows indicate the phase relationship, with in-phase/anti-phase relation shown by arrows pointing to the right/left, while downward (upward) direction means that GTEC is leading (lagging). X-axis labels refer to January 1st of the respective year.

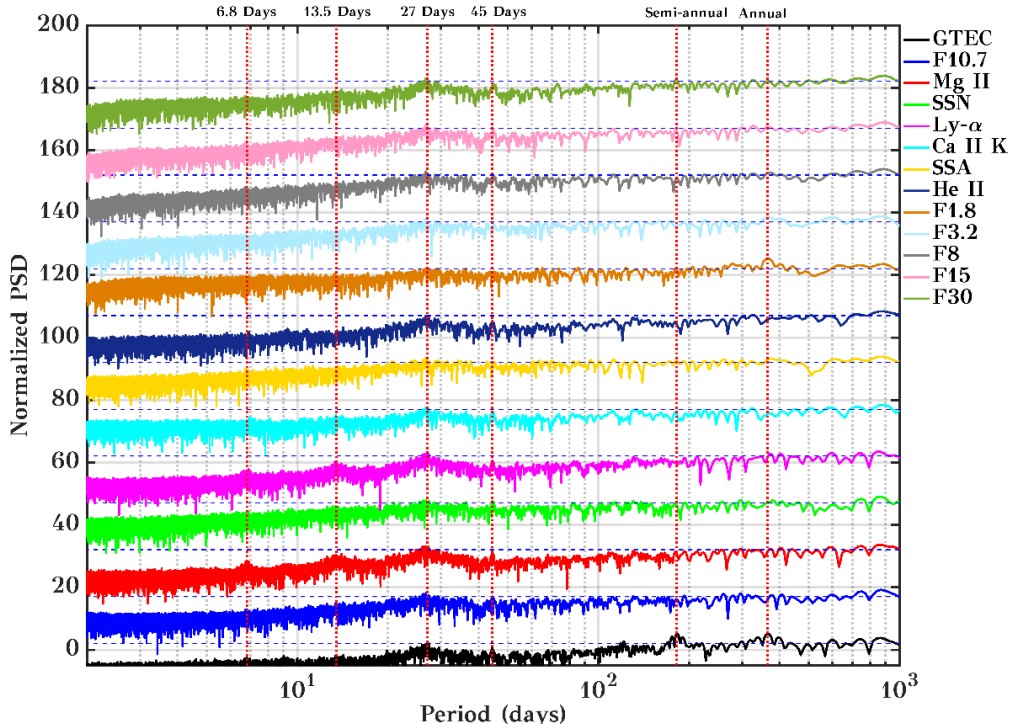

Figure 4: Lomb Scargle periodogram for GTEC and multiple solar proxies with a 95% confidence line (dashed blue color line). The curves are vertically offset by a factor of 15 each.

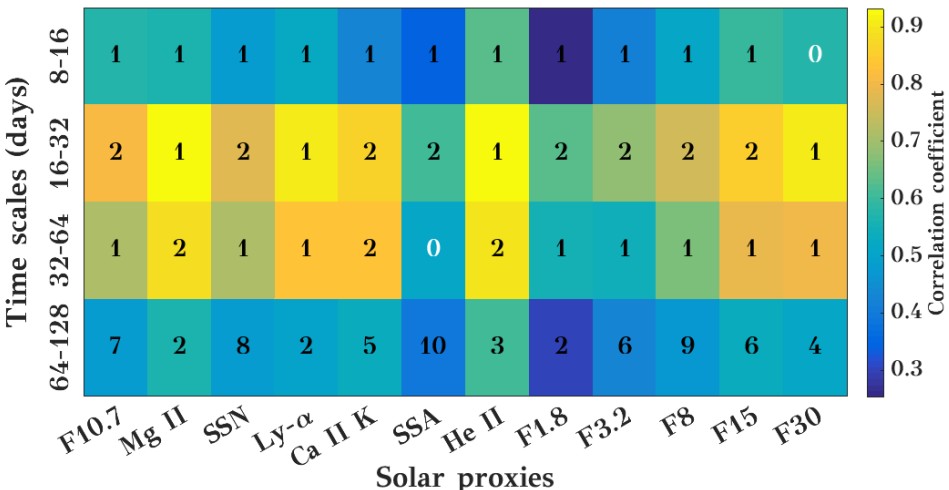

Figure 5: Wavelet cross-correlation sequence estimates for the maximal overlap discrete wavelet transform for GTEC and multiple solar proxies for different time scales (8 to 16, 16 to 32, 32 to 64, and 64 to 128-day). The background color shows the correlation coefficient, and the inserted number shows the delay in days.

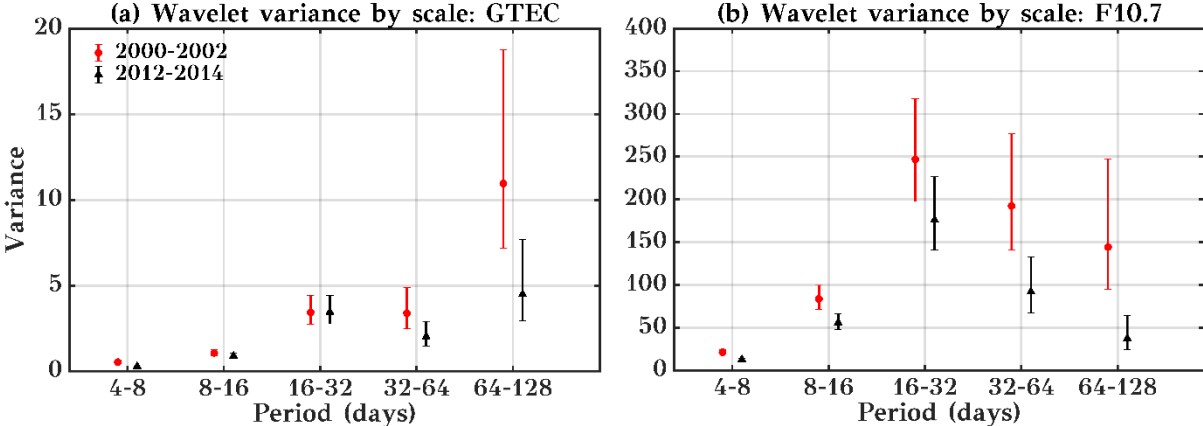

Figure 6: Wavelet variance for the maximums of SC 23 (2000-2002, red) and 24 (2012-2014, black) for (a) GTEC and (b) F10.7. Error bars show the 95% coverage probability of the confidence interval obtained from the 'Chi2Eta3' confidence method.

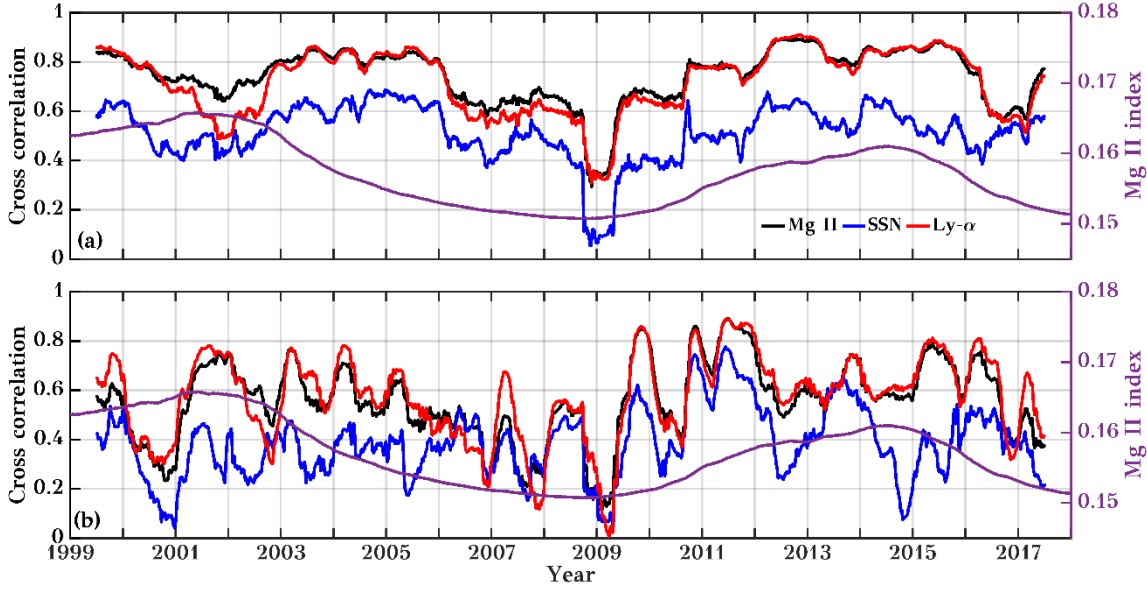

Figure 7: Running cross-correlation between GTEC and different solar proxies for (a) short (27 days residual), and (b) intra-annual time scales (original time series). For the short time scale the 27 days residual have been calculated by removing 27 days running mean from the original datasets. A 365 days running window is used to calculate the correlation. The second y-axis shows 365 days (upper and lower panel) running mean time series of Mg II index. Here Mg II, SSN, and Ly-α is marked by black, blue, and red colors, respectively.

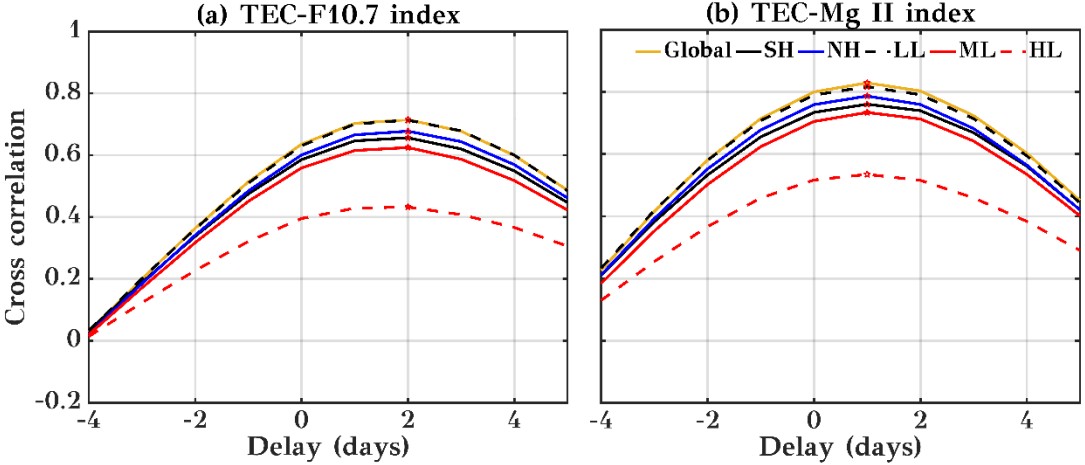

Figure 8: Cross-correlation coefficients and time delays between the Global, northern hemisphere (NH), southern hemisphere (SH) as well as low (LL, ±30°), middle (ML, ± (30°-60°)) and high (HL, ± (60°-90°)) latitude TEC with (a) F10.7 and (b) Mg II index during the years 1999 to 2017 for different lag. A positive lag means that solar flux variations are heading TEC ones. The maximum correlation is indicated by a red star.

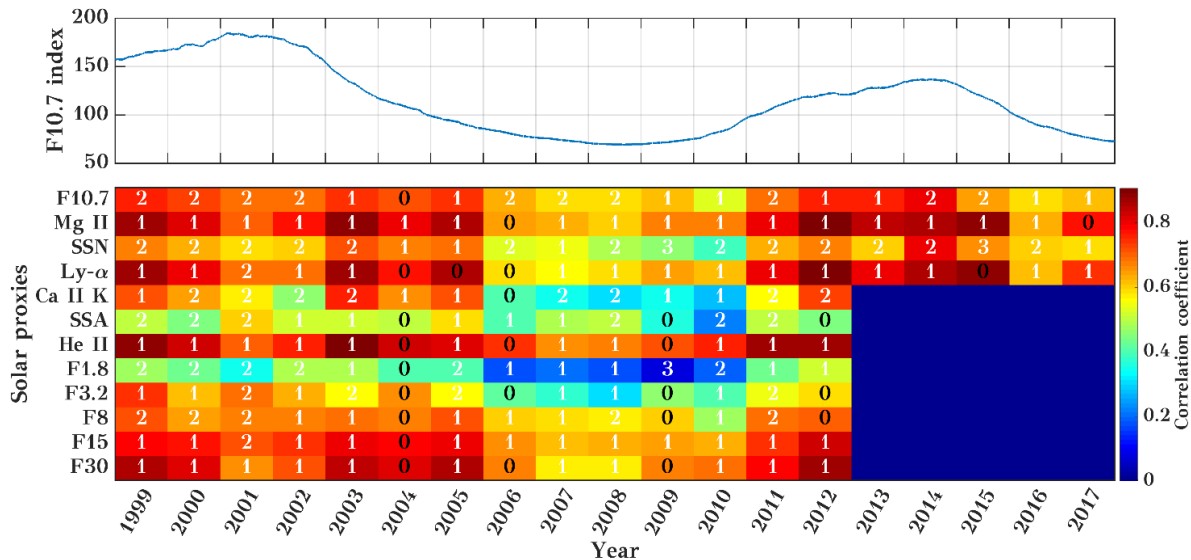

Figure 9: In the upper panel, 365 days running mean time series of F10.7 is shown. The lower panel displays yearly cross-correlations and time delays between GTEC and different solar proxies for the years 1999 to 2017 at the timescale of 16 to 32-day. The background colors give the maximum correlation coefficient, and the inserted numbers show the delay in days corresponding to maximum correlation.

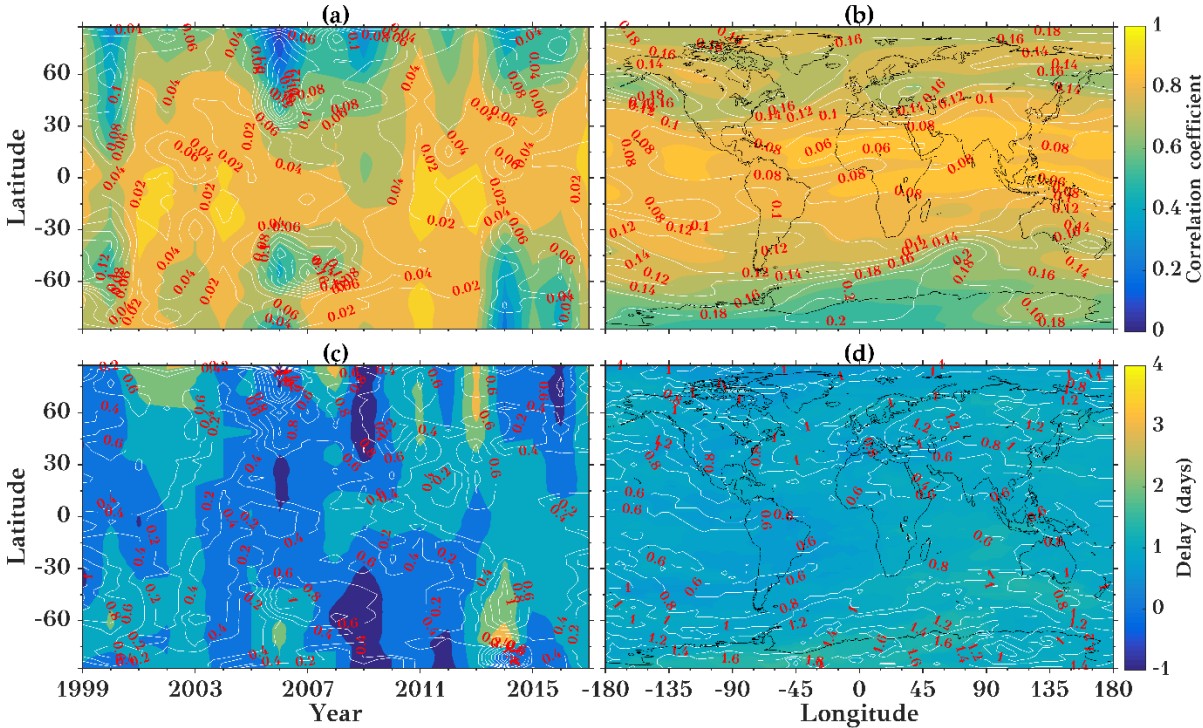

Figure 10: (a,c) Zonal mean and (b,d) long-term mean correlation coefficients (upper row) and time delay (lower row) between TEC and Mg II index for the years 1999 to 2017. The white contour lines indicate the respective standard deviations.

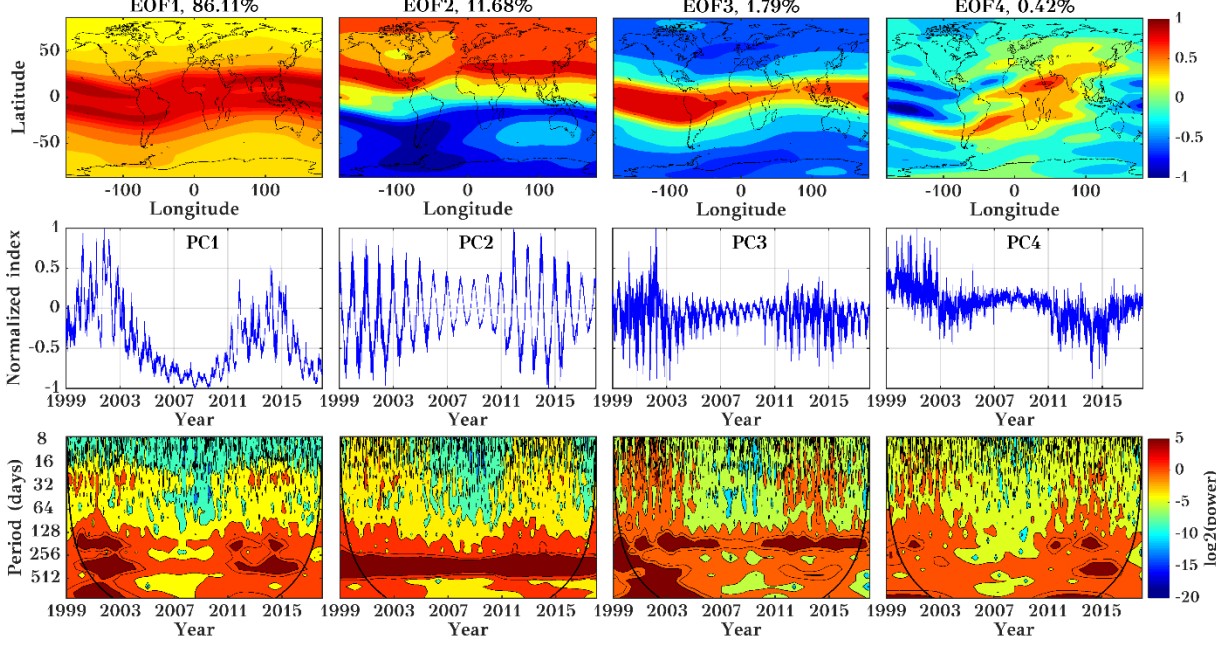

Figure 11: The first four EOFs (top row) of normalized TEC during the years 1999 to 2017, corresponding principal components (middle row), and their corresponding wavelet transform (bottom row, wavelet power in log2 scale). Please note that EOFs are dimensionless.

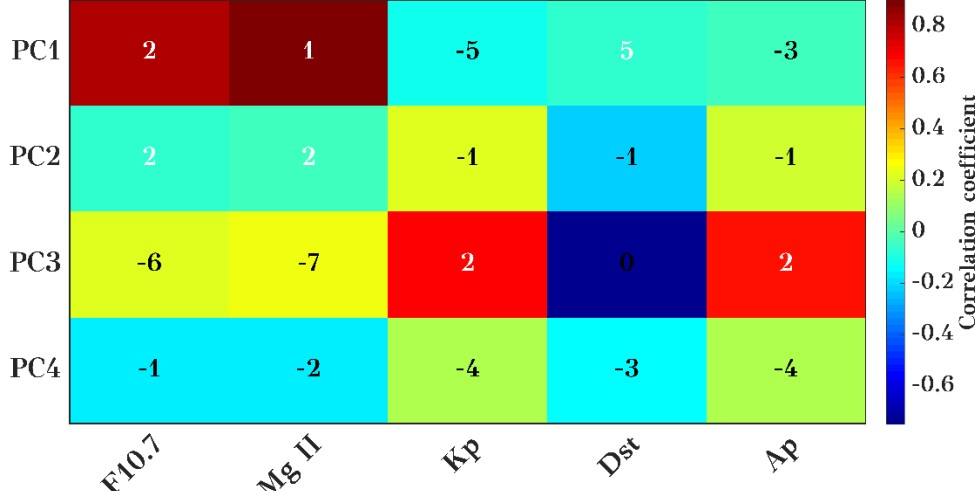

Figure 12: Correlation coefficients and time lag between the PCs and solar and geomagnetic proxies. Background colors show the maximum correlation coefficients, and the inserted numbers show the delay in days corresponding to maximum correlation.

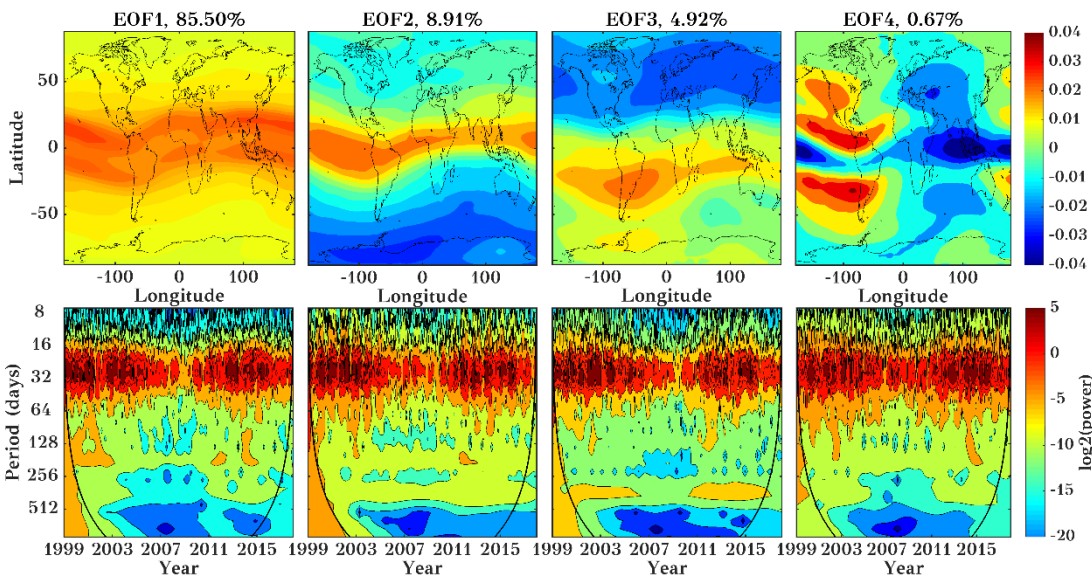

Figure 13: Spatial distribution map of first four EOFs (upper panel) of IGS TEC during the years 1999 to 2017, and their corresponding wavelet transform (lower panel, wavelet power in log2 scale) using a 25-35 days filtered datasets. The EOFs are dimensionless.

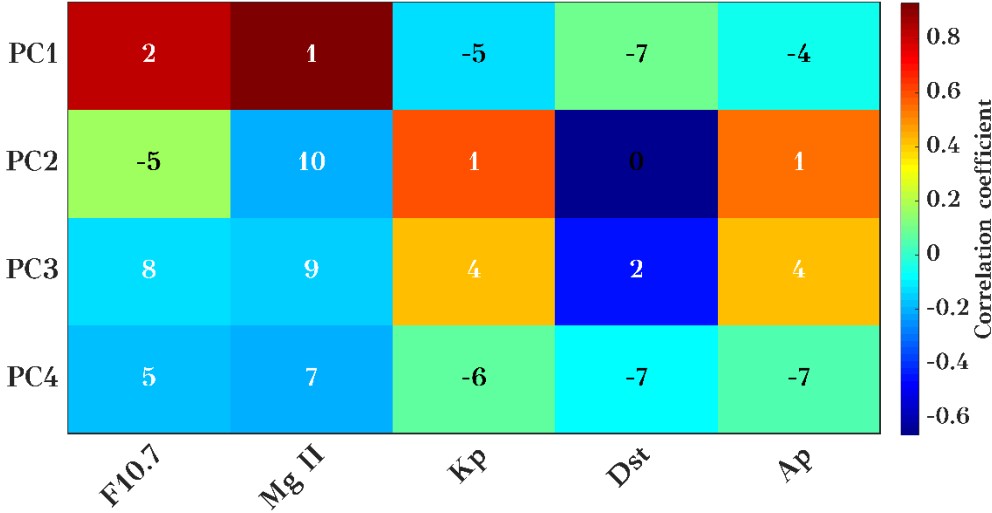

Figure 14: Maximum correlation coefficients and the time lag between the PCs and solar and magnetic proxies for the 25-35 days interval. Background colors show the maximum correlation coefficients, and the inserted numbers show the delay in days corresponding to maximum correlation.

695

## Appendix A: Additional figures

Additional figures are shown in order to complete the presentation. Figure A.1 is similar to Figure 5, but shows the correlation between solar proxies and GTEC at zero lag at different time scales. Figure A.2 shows the correlation at 16 to 32-day time scale between solar proxies and GTEC similar as Figure 9, but again at zero lag. 700 Figure A.3 is similar to Figure 9, but shows the cross-correlation and delay at the timescale of 32 to 64-day. Figure A.4 is similar to Figure 12 shows the correlation between PCs and solar and geomagnetic proxies, but at zero lag, while Figure A.5 shows the same for the 25-35 days interval. Running correlations at the intra-annual timescales, similar to Fig. 7 but also for different latitude ranges are shown in Fig. A.6. Fig. A.7 shows running cross-correlation between the PCs and different solar proxies and geomagnetic activity parameters.

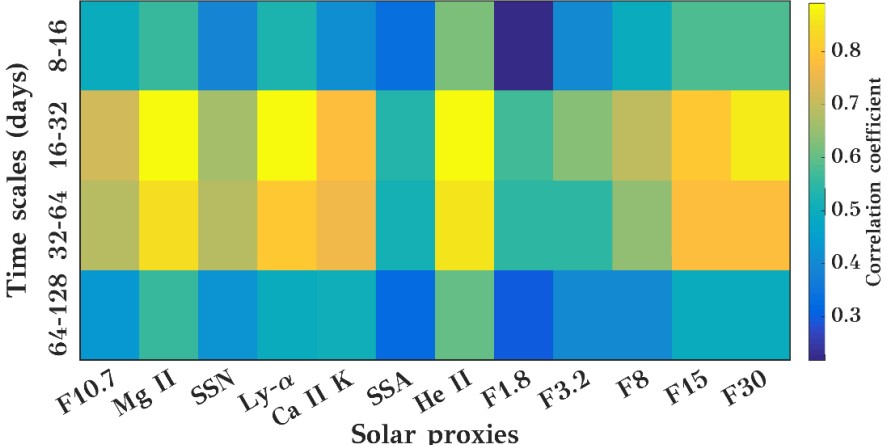

A.1: Wavelet cross-correlation sequence estimates for the maximal overlap discrete wavelet transform for GTEC and multiple solar proxies for different time scales (8 to 16, 16 to 32, 32 to 64, and 64 to 128-day). The background color shows the correlation coefficient at lag 0.

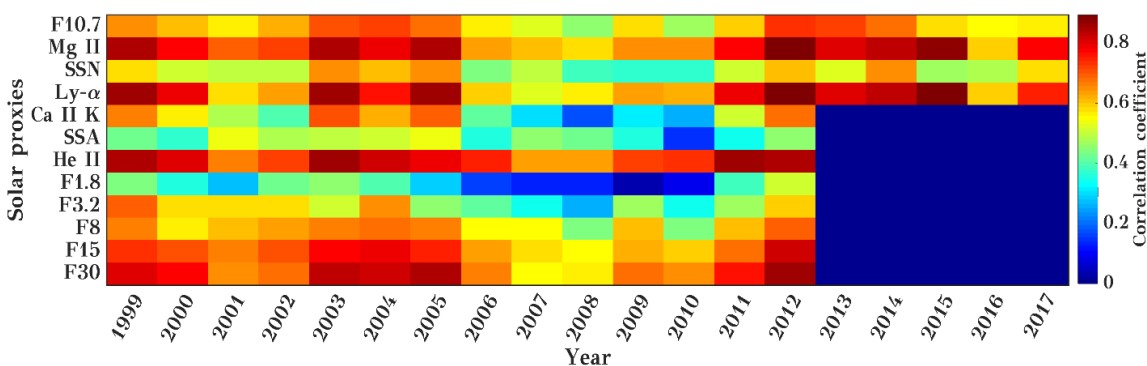

A.2: Cross-correlation between GTEC and different solar proxies for years 1999 to 2017 at the timescale of 16 to 32-day at lag 0. Background colors show the correlation coefficients.

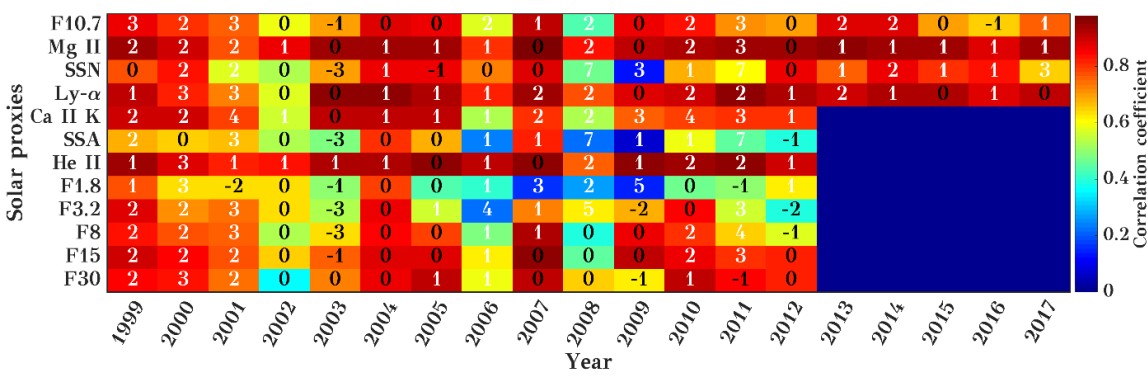

A.3: Cross-correlation and time delay between GTEC and different solar proxies for the years 1999 to 2017 at the timescale of 32 to 64-day. Background colors show the maximum correlation coefficients, and the inserted numbers show the delay in days corresponding to maximum correlation.

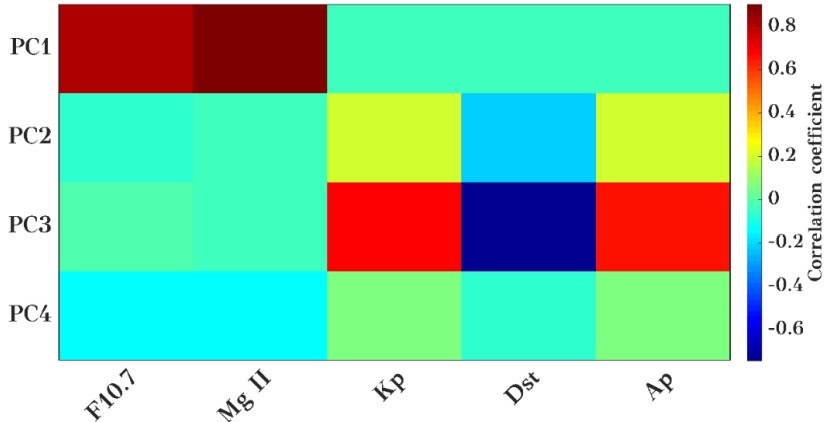

A.4: Correlation coefficients between the PCs and solar/geomagnetic proxies at lag 0.

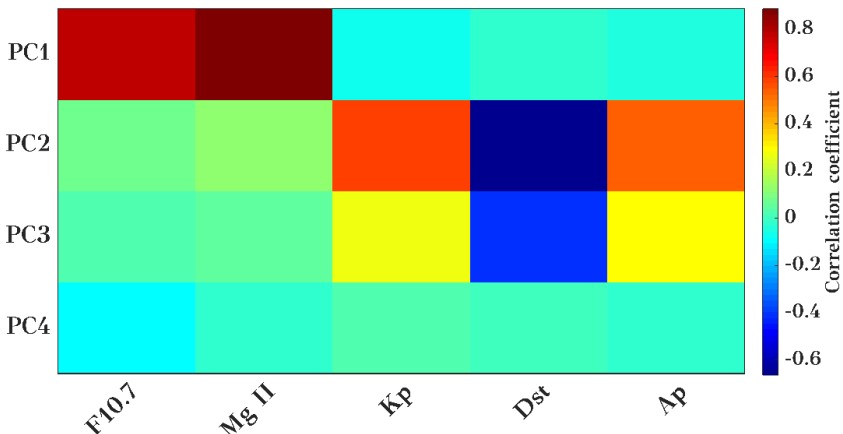

720      A.5: Correlation coefficients between the PCs and solar and magnetic proxies for the 25-35 days interval for zero lag.

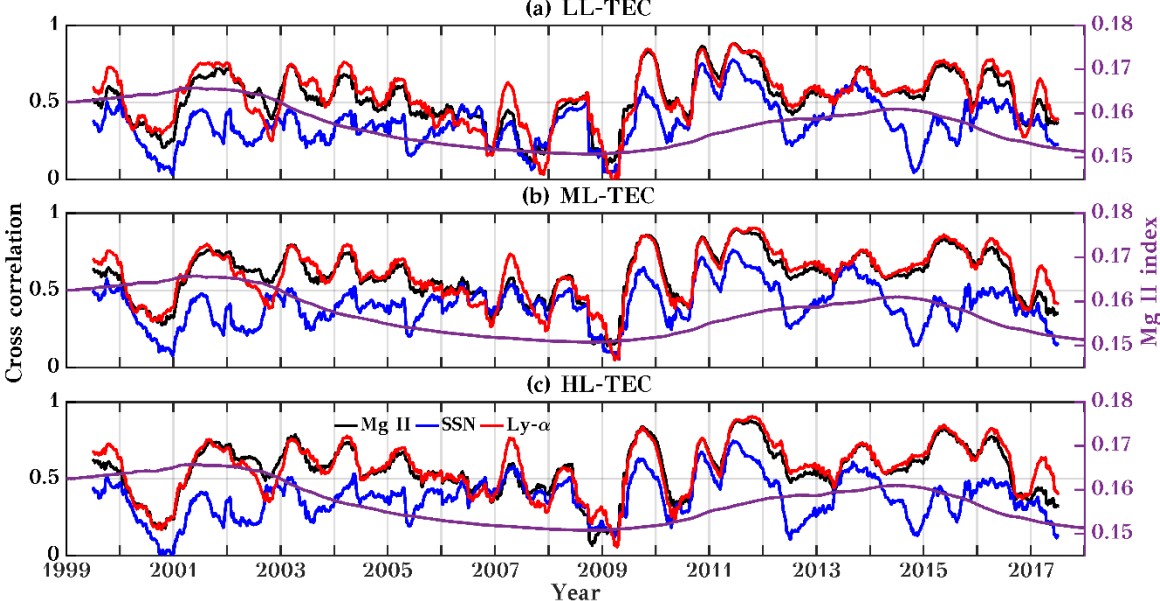

A.6: Running cross-correlation between the TEC and different solar proxies using a 365-day running window for LL, ML, and HL. The second y-axis shows the 365-day running mean time series of the Mg II index. Here Mg II, SSN, and Ly-α is marked by black, blue, and red colors, respectively.

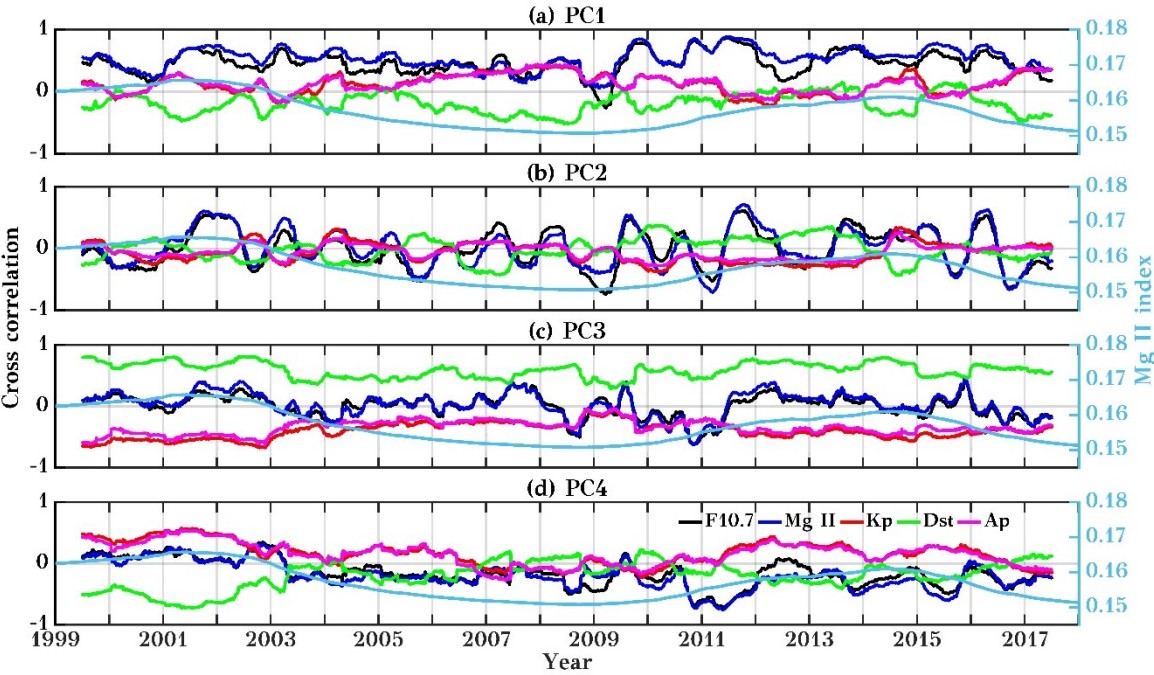

A.7: Running cross-correlation between the PCs and different solar proxies and geomagnetic activity parameters using a 365-day running window. The second y-axis shows the 365-day running mean time series of the Mg II index. Here F10.7, Mg II, Kp, Dst, and Ap are marked by black, blue, and red, green, and magenta colors, respectively.