# Peer review of "Long-term trends in the ionospheric response to solar EUV variations"

_Annales Geophysicae, 2019_

## Referee Comment (RC1) · Anonymous Referee #1 · 3 Apr 2019

The manuscript provides the investigation of the ionospheric response to the temporal and spatial dynamics of the solar activity by using 18 years solar activity indicators and also some geomagnetic activity indices. The topic is relevant and important for the community. In general, manuscript written good, but there are some problems in the manuscript. Authors need to consider these problems before resubmitting a revised version of the manuscript.

General comments about the manuscript

In Figure 1b and Figure 4 parameters do not separated easily, please use different colors as much as possible for each parameter. In the current version especially red and pink colors are mixing.

[Figure]

All abbreviations should be described clearly in the first place that they appear in the manuscript. In the current version of the manuscript some of them are not given with full name. Also, for the daily sunspot area the abbreviation is given as DSA. Please replace it as daily SSA

In Figure 4 the significance levels of obtained periodicities are not given. I suggest that authors should add at least 95 % confidence level line to each periodogram.

Please add some information about the appendix figures inside the manuscript.

Page 1 line 21, authors mentioned that "Wavelet variance estimation suggests that GTEC variance is highest for the seasonal timescale followed by the 16-32 days period, similar to the F10.7 index highest variance for the 16-32 days period." Please replace as "Wavelet variance estimation suggests that GTEC variance is highest for the seasonal timescale followed by the 16-32 days period, similar to the F10.7 index.

Line 25 "DSA" – "Daily SSA"

Line 34 "(e.g. Schmölter et al., 2018)", please add a few more reference.

Page 2 line 55, "…at different time scales." – "at different time scales such as (…)." Please clarify

Page 4 line 136 "…GTEC with four selected solar proxies…" please give these solar proxies inside a parenthesis.

In page 5 line 157, authors mentioned that they used 7 days smoothed data and they mentioned 6.7 days periodicity. From 7 days smoothed data it is not possible to get 6.7 days periodicity. This part should be removed.

Authors mentioned 128 – 256 days periodicity from GTEC and solar parameters. Source of this periodicity should be given more clearly (see Lou et al. 2003, Kilcik et al, 2018). For the 45 days periodicity, it is also one of the fundamental periodicity of solar activity and it detected in many solar activity indices (Lou et al. 2003, Chowdhury

et al. 2015, Kilcik et al, 2018). Please explain this periodicity a bit more detail. (Lou, Y.Q., Wang, Y.M., Fan, Z., Wang, J.X., Wang, S.: 2003,Mon. Not. Roy. Astron. Soc. 345, 809. Chowdhury, P., Choudhary, D.P., Gosain, S., Moon, Y.J.: 2015, Astrophys. Space Sci. 356, 7. Kilcik, A., Yurchyshyn, V., Donmez, B., Obridko, V.N., Ozguc, A., Rozelot, J.P.: 2018, Solar Phys. 293, 63.)

In page 6 line 179, authors mentioned that "...solar rotation period of 27 days is only a mean value and different solar regions rotate with a different velocity which can be up to 35 days." Please replace this sentence as "...the 27 days periodicity is only a mean value of solar differential rotation. It also strongly depends on the life time and proper motion of observed active regions."

Page 6 line 204, "The correlation coefficient is also decreasing during high solar activity years such as 2002 and 2014 but increases during the recovery phase of solar activity." This sentence is not correct, it should be clarified.

Page 8 line 246, authors mention that "The F1.8 and DSA cannot adequately represent the solar activity at the solar rotation (16-32 days) time scale." SSA is one of the best solar indicator in solar physics literature, so please clarify this sentence with more detail.

In line 264, "...several other physical processes." Please clarify these processes

In general, please use wavelet scalogram instead of wavelet transforms for wavelet plots. Also in the wavelet plots, what is the meaning of negative power it should be explained clearly or wavelet scalograms should be modified.

I think current version of the manuscript is not appropriate for the publication in the journal. It needs some corrections.

---

## Short Comment (SC1) · 16 Apr 2019

My comments:

1. You are using global TEC from GIM maps. However, there is a jump in GIM TEC in 2001, the values before being lower than values after (see Emmert et al., JGR-SP, 2017, doi:10.1002/ 2016JA023680). It has probably no effect on your result but for your future work.

2. The 27-day variation in the lower ionosphere (D-region) is often predomianntly caused by dynamical forcing (PW), not by direct solar forcing, particularly in winter unde rlow solar activity (Pancheva et al., JATP, 1991, https://doi.org/10.1016/0021-9169(91)90064-E).

---

## Author Comment (AC1) · 2 May 2019

Answer to Reviewer #1:

We are thankful for the reviewer's comments and suggestions which help us to improve the quality of the manuscript. We will address all the raised points in the revised version of the manuscript.

General comments about the manuscript

In Figure 1b and Figure 4 parameters do not separated easily, please use different colors as much as possible for each parameter. In the current version especially red and pink colors are mixing.
Response: The Figures will be modified in the revised version of the manuscript.

All abbreviations should be described clearly in the first place that they appear in the manuscript. In the current version of the manuscript some of them are not given with full name. Also, for the daily sunspot area the abbreviation is given as DSA. Please replace it as daily SSA
Response: We will add the descriptions of abbreviation and replace DSA with SSA in the revised version.

In Figure 4 the significance levels of obtained periodicities are not given. I suggest that authors should add at least 95 % confidence level line to each periodogram.
Response: We will add this in the revised version.

Please add some information about the appendix figures inside the manuscript.
Response: We will add the description of appendix figures in the revised version.

Page 1 line 21, authors mentioned that "Wavelet variance estimation suggests that GTEC variance is highest for the seasonal timescale followed by the 16-32 days period, similar to the F10.7 index highest variance for the 16-32 days period." Please replace as "Wavelet variance estimation suggests that GTEC variance is highest for the seasonal timescale followed by the 16-32 days period, similar to the F10.7 index.
Response: We will replace this sentence as suggested.

Line 25 "DSA" – "Daily SSA"
Response: We will replace the word in the revised version.

Line 34 "(e.g. Schmölter et al., 2018)", please add a few more reference.
Response: We will add more references in the revised version.

Page 2 line 55, ": : :at different time scales." – "at different time scales such as (: : :)." Please clarify
Response: We apologize for the typo error. We will correct this in the revised manuscript.
"Hocke (2008) studied oscillations in the global mean TEC (GTEC) and solar EUV (MG-II index) and reported dominant periods of solar rotation, annual, semi-annual, and solar cycle. These oscillations observed in GTEC could be related to the ionising radiation changes."

Page 4 line 136 ": : :GTEC with four selected solar proxies: : :" please give these solar proxies inside a parenthesis.
Response: We will add these proxies as suggested.

In page 5 line 157, authors mentioned that they used 7 days smoothed data and they mentioned 6.7 days periodicity. From 7 days smoothed data it is not possible to get 6.7 days periodicity. This part should be removed.
Response: We will remove this part of the sentence in the revised version.

Authors mentioned 128 – 256 days periodicity from GTEC and solar parameters. Source of this periodicity should be given more clearly (see Lou et al. 2003, Kilcik et al, 2018). For the 45 days periodicity, it is also one of the fundamental periodicity of solar activity and it detected in many solar activity indices (Lou et al. 2003, Chowdhury et al. 2015, Kilcik et al, 2018).
Please explain this periodicity a bit more detail.
(Lou, Y.Q., Wang, Y.M., Fan, Z., Wang, J.X., Wang, S.: 2003, Mon. Not. Roy. Astron. Soc. 345, 809.
Chowdhury, P., Choudhary, D.P., Gosain, S., Moon, Y.J.: 2015, Astrophys. Space Sci. 356, 7.
Kilcik, A., Yurchyshyn, V., Donmez, B., Obridko, V.N., Ozguc, A., Rozelot, J.P.: 2018, Solar Phys. 293, 63.)
Response: Thank you for the suggestion. We will add a description of the sources of periodicities in the revised manuscript.

In page 6 line 179, authors mentioned that ": : :solar rotation period of 27 days is only a mean value and different solar regions rotate with a different velocity which can be up to 35 days." Please replace this sentence as ": : :the 27 days periodicity is only a mean value of solar differential rotation. It also strongly depends on the life time and proper motion of observed active regions."
Response: Thank you for the suggestion. We will replace this sentence in the revised version.

Page 6 line 204, "The correlation coefficient is also decreasing during high solar activity years such as 2002 and 2014 but increases during the recovery phase of solar activity." This sentence is not correct, it should be clarified.

Response: We agree with the reviewer's point of view, and we will improve the description in the revised manuscript and add modified figures for short, interannual and longer time scales to explain the behaviour at different time scales.

Page 8 line 246, authors mention that "The F1.8 and DSA cannot adequately represent the solar activity at the solar rotation (16-32 days) time scale." SSA is one of the best solar indicator in solar physics literature, so please clarify this sentence with more detail.

Response: We agree with the reviewer concern. Most of the solar proxies (e.g., SSN, CaK, F10.7, Mg-II index) are the best solar indicator at longer time scales (e.g. solar cycle) but poorly correlated at short time scales (e.g., daily, solar rotation period). At longer time scale solar EUV and solar proxies are mainly controlled by solar magnetic activity. However, at short time scale, it varies differently as they originate from different excitations mechanism. Hence at the 16-32 days time scale, most of the solar proxies are weakly correlated with the solar EUV and as a result there is less correlation with GTEC. Hence in comparison to other solar proxies, F1.8 and SSA are poorly correlated with the GTEC. We will add short, longer and interannual time scales in figure 7 for a detailed explanation of solar proxies behaviour at different time scale. We will add a detailed description in the revised manuscript.

In line 264, ": : :several other physical processes." Please clarify these processes

Response: Ionospheric variability is strongly depending on the solar activity as well as geomagnetic and meteorological activity. So, the variability in the ionosphere is not only controlled by solar activity. During the low solar activity period, lower atmospheric forcing is more dominant. We will add these processes in the revised version.

In general, please use wavelet scalogram instead of wavelet transforms for wavelet plots. Also in the wavelet plots, what is the meaning of negative power it should be explained clearly or wavelet scalograms should be modified.

Response: Thank you for this suggestion. We will add the description of negative power in the revised version of the manuscript. To get clear periodicity from the wavelet, we have used log2 of (power). The negative (positive) values indicate the low(high) power. As there is no difference in periodicity estimation either we use transform or scalogram, so we will keep it the same in the revised version.

I think current version of the manuscript is not appropriate for the publication in the journal. It needs some corrections.

Response: Thank you for reviewing our manuscript. We will address all the comments in the revised version of the manuscript.

---

## Author Comment (AC2) · 2 May 2019

Answer to Reviewer:

We are thankful for the reviewer's suggestions which help us to improve the quality of the manuscript.

1. You are using global TEC from GIM maps. However, there is a jump in GIM TEC in 2001, the values before being lower than values after (see Emmert et al., JGR-SP, 2017, doi:10.1002/ 2016JA023680). It has probably no effect on your result but for your future work.
Response: Thank you for the suggestion. Yes, we can see a clear jump in GIM TEC in 2001. We will discuss this in the revised version.

2. The 27-day variation in the lower ionosphere (D-region) is often predominantly caused by dynamical forcing (PW), not by direct solar forcing, particularly in winter under low solar activity (Pancheva et al., JATP, 1991, https://doi.org/10.1016/0021-9169(91)90064-E).

Response: Thank you for the suggestion and we will discuss this in the revised version of the manuscript. However, D region ionization contributes only weakly to TEC.

---

## Referee Comment (RC2) · Anonymous Referee #2 · 29 May 2019

The manuscript presents a new and very interesting study of the ionospheric response to solar activity. Solar activity is represented by individual solar proxy datasets. The authors study the correlation and lag of the variability of solar proxies with the response in the ionosphere/thermosphere, represented by the total electron. Of key interest is which solar activity proxies best describe the ionospheric response. In their study the authors employ a principal component analysis, empirical orthogonal functions (EOFs) as well as the cross-wavelet analysis and Lomb Scargle periodogram (LSP).

Major Comments: =============== The authors present new and also very interesting results. However, additional clarifications are still to recommend the paper for publication.

The result of the lag is presented in the text mainly as lag of one or two days. As the

result of the lag is also an important result it needs to be presented more precisely, i.e. with at least one (or better two) digits after the comma, i.e. 1.8 (e.g. from Figure 8), corresponding to maximum of the cc-curve?

In the abstract (Line 26ff) the authors state that "Empirical orthogonal function (EOF) analysis of the TEC data shows that the first EOF components capture more than 86% of the variance, and the first three EOF components explain 99% of the total variance." The authors should specify who the contribtuions (86%, ect) are determined. The authors state that the first EOF is the solar component. Could the authors elaborate on the other EOFs under consideration (in particular the 2nd and 3rd). This is partly done in lines 192ff. Could the dynamics of the Earth's atmosphere also play a role?

The link between the EOFs and PCs is not clear. In Figure 11 the authors plot EOF1 to EOF4. In Figure 12 and 14 the authors show the CC of the PCs with proxies, and in Lines 302 the authors state "In order to check the relation between solar proxies and geomagnetic parameters (daily Kp, Dst, and Ap indices) with PCs corresponding to EOFs, cross-correlation and delay is calculated and shown in Figure 12." Could the authors elaborate better how the EOFs and PCs are derived and what is the time series for the CC in Figs 12 and 14.

In the introduction, further references to previous work should be mentioned e.g.: http://adsabs.harvard.edu/abs/2016JGRA..12110367L and others.

For the determination of the lag is not clear. How is it derived. Possibly it should be the lag value for the maximum correlation. Please give precise values for the lag (e.g. 1.8 in Figure 8.)

Minor comments: =============== Line 2: please clarify or rephrase "spatial dynamic of solar activity". The solar proxies under consideration do contain any spatial information, possibly the authors mean "the spatial response of the ionosphere to sola activity"? Line 10: GNSS, explain acronym when first mentioned Line 35ff: "These studies have shown, that the response of the ionosphere to solar EUV radiation variations takes 1-2 days for solar radiation changes within 27 days solar rotation period". This sentence is not clear, could the authors please rewrite it.

Line 13: A 16-32 days period -> A 16 to 32-day period (day, without s) Line 15: LSP analysis -> The LSP analysis Line 18: "The wavelet variance estimation method is used to find the variance in the maximum of the solar cycles (SC) 23 (2000-2002) and 24 (2012-2014), for GTEC and F10.7 index, respectively. " Suggested rephrasing, as the sentence does not read very well. -> The wavelet variance estimation method is used to find the variance of GTEC and F10.7 over the maxima of the solar cycles SC 23 and SC 24. The selected time frame that covers the solar maxima are .... and ....

Line 20: GTEC variance -> the GTEC variance Line 20: seasonal timescale: which one is considered as the seasonal time scale? 32-64-day period? please specify or rather give the name of the wavelet window. Generally, the wavelet intervals could be numbered so that the intervall does not need to be repeated in the text again.

Line 22: to represent the solar activity -> to represent solar activity Line 23: may be placed at the second ... -> may be placed second ... Line 24: but there are some differences between solar maximum and minimum: could the authors be more specific. Line 25: The F1.8 and DSA ... -> The indices F1.8 and DSA ... Line 26: Empirical orthogonal function (EOF) analysis -> The Empirical orthogonal function (EOF)

Line 27: "EOF analysis suggests that the first component is associated with the solar flux." This result is expected, but also very nice to be an outcome of the EOF analysis. Could the authors also indicate what the status of the knowledge/hypothesis about the nature of the subsequent 2-3 EOF components are (dynamics, ect). Line 33: reference Chen et al., 2012: Please add more references. Line 36 (and elsewhere in the manuscript): These studies have shown, that the response of the ionosphere to solar EUV radiation variations takes 1-2 days.: A quantitative analysis of the response time of the ionosphere to the EUV radiation is an important result. As already stated above, this needs to be presented in a more quantitative and presice way. Could the

authors also give the precise values for the lag for all studies undertaken, e.g. in a table, or in . Line 43: investigate ... mechanism -> investigate the ... mechanism Line 48: "The T-I system is also influenced by different external forces": the solar forcing should also be considered as "external forcing". Therefore, aren't all forcings "external"? Line 49: "In the case of solar events, the forcing from above might even result in strong disturbances affecting the ionospheric delay." -> This sentence needs to be revised. Suggestion: In addition to the solar EUV forcing, the solar wind as well as solar eruptions might also result in ... Could the authors give references that address this work? Line 50: "As a result, the ionospheric plasma behaviour is varying during different solar activity conditions." It is not clear what is meant here. Please revise this sentence. Line 51-58 (full paragraph): The authors mention the 27-day solar rotation period and its effects on the TEC. What is missing in Hocke (2008). Why are further investigations needed? Line 57ff: "Many studies ...". Sentence seems out of place here, move above as the paragraph above seems to be the introduction to the 27-day variability. Also please give some references to the "many studies". Line 59: Since direct EUV measurements ... and are still not available in the full spectrum...: In recent times the situation of the EUV measurements has considerably improved (thanks to e.g. SDO/EVE, see also http://lasp.colorado.edu/lisird/). Also, while degradation of space instruments is still a challenge, the availability of SSI data in the EUV (either direct measurements, composite datasets or models) has improved, see e.g. Lean et al. http://adsabs.harvard.edu/abs/2003JGRA..108.1059L Haberreiter et al., 2017, composite covers the full spectrum, incl. the EUV Please revisit the statement. Line 63: .. and indices based on direct EUV measurements (e.g., Unglaub et al., 2011) like the Solar EUV Experiment (SEE) onboard the Thermosphere Ionosphere Mesosphere Energetics and Dynamics (TIMED) satellite (Woods et al., 2000). -> .. and indices developed by Unglaub et al. (2011) based on direct EUV measurements obtained with the Solar EUV Experiment (SEE) onboard the Thermosphere Ionosphere Mesosphere Energetics and Dynamics (TIMED) satellite (Woods et al., 2000). Line 65: which may be overcome by repeated calibration -> please clarify what is meant here,

inflight calibration is repeated calibration. Do the authors mean rocket calibrations of flight spares as done with SDO/EVE? Line 93: OMNIWeb Plus database: please give a reference and/or link to the database. Line 103: The zonal mean plot additional temporal variations: please explain which those are. Line 105: around the magnetic equator: the variation seems rather symmetric around the equator. The magnetic equator should possibly be indicated in the plot if possible, or would "the equator" be sufficient. Line 107: is varies -> varies Line 109: delete "E.g." at the beginning of the sentence Line 110: "As all the time series in Figure 1 show a similar overall variation during the 11-year solar cycle, the fundamental behaviour of solar radiation emission is identical at all the wavelengths." A lot of care needs to be taken here. Actually, the fundamental behaviour is not the same for all wavelength, as the plasma heating and atomic processes are different for different wavelength. Specifically, for the radio proxies the processes for the various proxies are different (see Dudok de Wit, et al., 2014, http://adsabs.harvard.edu/abs/2014JSWSC...4A..06D) Line 120: Note that the T-I system is not only influenced by solar activity but also by changing geomagnetic conditions due to solar wind variations. -> Please revise, suggestion: Note that the T-I system is not only influenced by the solar electromagnetic radiation but also by changing solar energetic particles and geomagnetic conditions due to solar wind variations or Coronal Mass Ejections reaching the Earth. Please also give reference to support this. Effect of particles on the Earth upper atmosphere? Line 121: Strong solar activity during solar maxima might induce stronger interaction...: Please revise sentence, suggestion: The response to solar forcing is higher during solar maximum... Please also add references. The solar wind, in particular from coronal holes, also occurs during solar minimum conditions. Please take this also into account. Line 135: This allows to determine dominant joint oscillations -> This allows us to determine dominant correlated oscillations (or other word for "joint") Line 137: 16-32 days period region -> 16 to 32-day interval (also elsewhere in the text) Line 138: the ionospheric variation due to the solar activity is lower -> the ionospheric variation is lower due to solar activity Line 142: The black arrows in Figure 3 indicate the phase relationship between solar proxies

and GTEC (also caption of Figure 3): What does the upward orientation of the arrow mean? Line 144: the annual and semi-annual period range -> ranges? Could you give the exact interval for those. It is two separate intervals that are meant here? Please clarify. Line 153: ... semi-annual. The observed periodicities in GTEC are also shown by Hocke (2008). -> ... semi-annual, which is in line with Hocke (2008) (if this is what is meant). Line 154: It is interesting to note here that a 44-day periodicity is observed in GTEC and all other solar proxies.: From Figure 4 the 44-day variability seems not significant. It seems that there is random variability in the window up to 1/2 year. of the same order of magnitude in the time series. Without further analysis it cannot be stated that a 44-day variability is visible in "all other solar proxies". Please revise. Line 156: .. and it's 2nd harmonic 13.5 days, and 4th harmonic 6.7 days...: Please also indicate these harmonics in Figure 17. Line 157: Here similar kind of oscillations..: Do the authors find the same oscillations, i.e. 2nd harmonic 13.5 days, and 4th harmonic 6.7 days. Or are they different for Lyman alpha. If so, please specify. Line 157: Ly-$\alpha$ - Ly-$\alpha$ (take out space) Line 159ff: Note that the wavelet spectra show some periodicity at the half-year time scale, but with variable phase so that they extinguish in the periodogram.: This sentence needs to be revisited. For which proxies? In Figure 4 only the GTEC and and maybe F30 show a 1/2 year peak. Please be specific. Line 162: Maybe add a subtitle here: "Wavelet Cross-Correlation" Line 164: using -> based on (repetition from line 163) Line 167: The delay is mostly positive or zero, which means that TEC is following the solar proxies with delay. -> The delay is mostly positive or zero, which means that TEC is following the solar proxies. Line 170: ... by about one day: could you please give the exact value here (and everywhere in the text when the lag is given)? E.g. Line 176, 176 Line 184: A stronger correlation -> A strong correlation Line 186: for the GTEC -> for GTEC Line 186: with Daubechies 2 ... -> with the Daubechies Line 194: There is no strong semi-annual cycle visible. -> ... and as expected, no significant semi-annual cycle is visible. Line 199: inter-annual time scales: these are timescales of one year or larger? Please clarify. Maybe "time scales below (or above) one year" Line 200: 365 days running window -> 365-day running winding

Line 203: All solar proxies show similar behaviour during low activity conditions: While the temporal variation of the CC for Mg II, Ly alpha and He II is largely similar, the SSN (green curve) shows a significantly different behaviour. Line 203: apart from a different mean level: Not sure what the author mean as "mean level". It could be stated that SSN generally shows a lower CC than the other proxies. Line 208: Are the cross correlations shown in Fig. 8 a temporal mean over the years 199 to 2017. It would be very interesting to see the temporal variation, e.g. similar to Fig. 7, if possible. Line 210ff: "As in Figure 7, the correlation of F10.7 with TEC is weaker than the one of MG-II and TEC.". However, F10.7 is not shown in Fig. 7. Please revise. Line 210ff (disucussion of Fig. 8). Maybe start out with: Generally, the correlation coefficient and the lag for the Global, NH, SH, LL, and ML are very close. Then continue: The maximum correlation is found... The weakest correlation is observed... Line 211: HL with maximum correlation coefficients -> HL with a maximum correlation coefficient of Line 215: "response time of about two and one days": As already mentioned above, could the authors give a more precise result for the lag, i.e. derived from the maximum of the curves in Fig. 8. It might be something like 1.8 or 1.9. Please also mark the maximum of each curves in Fig. 8 with a cross or similar. Line 225: is shown -> is found Line 235: From the above discussion it is clear that during -> In summary, during low solar activity... ; please also add a summary sentence for high solar activity Line 238: indices shows stronger -> indices show a stronger Line 239: of about 1-3 days: again, please provide a more precise determinination of the lag. 274: using Empirical Orthogonal Function (EOF) which decomposes -> Empirical Orthogonal Functions (EOFs), which decompose Line 276: to represents -> to represent Line 282: How do the authors derive the contributions of the PC1, PC2, ect of about 86%, 11%, ect. Please add this to the text. The result of EOF2, 11%, is given in brackets in line 282 and 292 again. The first mention could be removed, as the text is then better to read. Line 299: remove "only", as both semi-annual and annual oscillations are visible. Lien 302: In order to check the relation between solar proxies and geomagnetic parameters (daily Kp, Dst, and Ap indices) with PCs corresponding to EOFs, cross-correlation and delay is calculated and shown in Figure 12. The color-coded value in Fig. 12 is the temporal average of the correlation coefficient? This should be stated in the text and the figure caption. Figure 12 (and Figure 14) are very interesting result of the paper. Therefore, it would be very interesting to also see the temporal variability of the correlation (also similar to Fig. 7 for the solar proxies). Could the authors provide this for completeness (at least for a few cases), possibly in the appendix? Line 315: is capturing -> captures

Figures: Figures in general: a number of panels are rather on the small side. It is recommended to fill the full text width in order to make the figures better readable Figure 1, 3, 8, 10, 11 : enlarge panels to 0.5 textwidth Figure 11: Panels need to be enlarged, as it is difficult to see the details in the printed version of the manuscript. The panels should be of the size of the new Fig. 10, i.e. only two panels next to each other. Could Figure 11 be split, say PC1 and PC2 in one figure, and PC3/PC4 in another? Caption Figure 8: please add: Temporal mean (cross correlation .... during the years 1999 to 2017) for different lags. Caption Figure 9: The background colors show the correlation coefficient -> The background colors give the temporal mean of the correlation coefficient (or maximum?), please clarify which cc is shown throughout the text.

Throughout the text: He-II-> He II (no hyphen) MG-II-> Mg II (no hyphen) CaK -> Ca II K 16-32 days period -> 16 to 32-day period (day always without s) thoughout the text and figure captions The authors often show a correlation coefficient (Figure 8, Figs 12, 14, ect). Please specify if it is a temporal mean, spatial mean, maximum value corresponding to a lag value? Use Figure xx (at the beginning of the sentence, as in Line 106) and its abbreviation Fig. xx (in sentence, as e.g. in line 105) consistently throughout the text. Generally, the second part of the paper, starting with lines 208 reads more fluent. Could the (co-)authors go over the manuscript for language and typos.

[Figure]

2019.

---

## Author Comment (AC3) · 13 Jun 2019

The manuscript presents a new and very interesting study of the ionospheric response to solar activity. Solar activity is represented by individual solar proxy datasets. The authors study the correlation and lag of the variability of solar proxies with the response in the ionosphere/thermosphere, represented by the total electron. Of key interest is which solar activity proxies best describe the ionospheric response. In their study the authors employ a principal component analysis, empirical orthogonal functions (EOFs) as well as the cross-wavelet analysis and Lomb Scargle periodogram (LSP).

Response: The authors are thankful to the reviewer for the critical comments and encouraging suggestions, which helped to improve the quality of the manuscript. We will address all the raised points in the revised version of the manuscript.

Major Comments: ================

The authors present new and also very interesting results. However, additional clarifications are still to recommend the paper for publication. The result of the lag is presented in the text mainly as lag of one or two days. As the result of the lag is also an important result it needs to be presented more precisely, i.e. with at least one (or better two) digits after the comma, i.e. 1.8 (e.g. from Figure 8), corresponding to maximum of the cc-curve?

Response: The authors are thankful to the reviewer for appreciating our findings. In this analysis, we have used all the datasets (GTEC and Solar proxies) at a daily resolution, as our used solar proxies are available in daily resolution. So, the calculated delay will be in the day(s) only. We will mark the maximum of cc curves in Fig. 8, where the delay is 1 or 2 days.

In the abstract (Line 26ff) the authors state that "Empirical orthogonal function (EOF) analysis of the TEC data shows that the first EOF components capture more than 86% of the variance, and the first three EOF components explain 99% of the total variance." The authors should specify who the contributions (86%,

ect) are determined. The authors state that the first EOF is the solar component. Could the authors elaborate on the other EOFs under consideration (in particular the 2nd and 3rd). This is partly done in lines 192ff. Could the dynamics of the Earth's atmosphere also play a role?

Response: In the revised version, we will add a brief description of EOF analysis in section 3.6, including the calculation of explained variance, and also we will update the abstract. Actually EOF1 includes the solar variability effect, as it is symmetric about the equator. This includes Earth's eccentricity effect also.
EOF2 is antisymmetric about the equator, with the maximum during the winter (see PC2). This indicated that EOF2 includes the seasonal effect, including dynamical effects in the ionosphere.
PC3 is correlated with magnetic activity (See Fig. 13) so we attribute EOF3 with magnetic effects. PC4 shows a slight trend. One may speculate that this might be due to secular changes of the magnetic equator.

The link between the EOFs and PCs is not clear. In Figure 11 the authors plot EOF1 to EOF4. In Figure 12 and 14 the authors show the CC of the PCs with proxies, and in Lines 302 the authors state "In order to check the relation between solar proxies and geomagnetic parameters (daily Kp, Dst, and Ap indices) with PCs corresponding to EOFs, cross-correlation and delay is calculated and shown in Figure 12." Could the authors elaborate better how the EOFs and PCs are derived and what is the time series for the CC in Figs 12 and 14.

Response: We will discuss the link between EOFs and PCs with the formula in section 3.6. To calculate the maximum CC in Figs 12 (new figure, 13) and Fig. 14 (new figure, 15), we have used time series of PCs and different parameters including solar and geomagnetic parameters from the 1 January 1999 to 31 December 2017.

In the introduction, further references to previous work should be mentioned e.g.: http://adsabs.harvard.edu/abs/2016JGRA..12110367L and others.
Response: We will add more references in the revised version.

For the determination of the lag is not clear. How is it derived. Possibly it should be the lag value for the maximum correlation. Please give precise values for the lag (e.g. 1.8 in Figure 8.)
Response: We have derived the lag as the value for the maximum correlation. We have used daily resolution datasets as all the used solar proxies are available in daily resolution. So, the calculated delay will be in the day(s) only.

Minor comments: =================

Line 2: please clarify or rephrase "spatial dynamic of solar activity". The solar proxies under consideration do contain any spatial information, possibly the authors mean "the spatial response of the ionosphere to solar activity"?
Response: We will rephrase this sentence as suggested.
.
Line 10: GNSS, explain acronym when first mentioned
Response: We will add the acronym.

Line 35ff: "These studies have shown, that the response of the ionosphere to solar EUV radiation variations takes 1-2 days for solar radiation changes within 27 days solar rotation period". This sentence is not clear, could the authors please rewrite it.
Response: We apologize for this error. We will correct this in the revised manuscript: "These studies have shown that the response of the ionosphere to solar EUV radiation variations gets delayed by 1-2 days at 27 days solar rotation period"

Line 13: A 16-32 days period -> A 16 to 32-day period (day, without s)
Response: We will improve this as suggested by the reviewer in the revised version.

Line 15: LSP analysis -> The LSP analysis
Response: We will improve this as suggested by the reviewer in the revised version.

Line 18: "The wavelet variance estimation method is used to find the variance in the maximum of the solar cycles (SC) 23 (2000-2002) and 24 (2012-2014), for GTEC and F10.7 index, respectively. " Suggested rephrasing, as the sentence does not read very well. -> The wavelet variance estimation method is used to find the variance of GTEC and F10.7 over the maxima of the solar cycles SC 23 and SC 24. The selected time frame that covers the solar maxima are .... and ....
Response: We will rephrase this sentence as suggested.

Line 20: GTEC variance -> the GTEC variance
Response: We will improve this as suggested by the reviewer in the revised version.

Line 20: seasonal timescale: which one is considered as the seasonal time scale? 32-64-day period? please specify or rather give the name of the wavelet window. Generally, the wavelet intervals could be numbered so that the interval does not need to be repeated in the text again.

Response: Yes, the 32 to 64-day period is considered as seasonal timescale. We will add this in the revised version of the manuscript.

Line 22: to represent the solar activity -> to represent solar activity
Response: We will improve this in the revised text.

Line 23: may be placed at the second ... -> may be placed second ...
Response: We will improve this in the revised text.

Line 24: but there are some differences between solar maximum and minimum: could the authors be more specific.
Response: We will rephrase the sentence in the revised version of the manuscript.

Line 25: The F1.8 and DSA ... -> The indices F1.8 and DSA ...
Response: We will improve this in the revised text.

Line 26: Empirical orthogonal function (EOF) analysis -> The Empirical orthogonal function (EOF)
Response: We will improve this in the revised text.

Line 27: "EOF analysis suggests that the first component is associated with the solar flux." This result is expected, but also very nice to be an outcome of the EOF analysis. Could the authors also indicate what the status of the knowledge/hypothesis about the nature of the subsequent 2-3 EOF components are (dynamics, ect).
Response: Thanks for the suggestion. We will add it in the abstract.

Line 33: reference Chen et al., 2012: Please add more references.
Response: We will add more references in the revised version.

Line 36 (and elsewhere in the manuscript): These studies have shown, that the response of the ionosphere to solar EUV radiation variations takes 1-2 days.: A quantitative analysis of the response time of the ionosphere to the EUV radiation is an important result. As already stated above, this needs to be presented in a more quantitative and presice way. Could the authors also give the precise values for the lag for all studies undertaken, e.g. in a table, or in.
Response: We completely agree with the reviewer's suggestion but the delay is still not analysed in high resolution due to unavailability of high-resolution datasets of solar EUV proxies. Schmölter et al. 2018 analysed SDO EVE and TEC datasets at hourly resolution and they reported a delay of about 17 hours. Hence due to unavailability of long-term high resolution solar EUV observations, in this paper, we have used daily datasets. Most of the researcher reported the delay about 1-2 days just because of the unavailability of hourly solar proxies.

Line 43: Investigate ... mechanism -> investigate the ... mechanism
Response: We will improve this in the revised text.

Line 48: "The T-I system is also influenced by different external forces": the solar forcing should also be considered as "external forcing". Therefore, aren't all forcings "external"?
Response: We agree with your concern. We will improve this in the revised manuscript.

Line 49: "In the case of solar events, the forcing from above might even result in strong disturbances affecting the ionospheric delay." -> This sentence needs to be revised. Suggestion: In addition to the solar EUV forcing, the solar wind as well as solar eruptions might also result in ... Could the authors give references that address this work?
Response: We will add references and rephrase the sentence as suggested.

Line 50: "As a result, the ionospheric plasma behaviour is varying during different solar activity conditions." It is not clear what is meant here. Please revise this sentence.
Response: We will revise it.

Line 51-58 (full paragraph): The authors mention the 27-day solar rotation period and its effects on the TEC. What is missing in Hocke (2008). Why are further investigations needed?
Response: We apologize for the typo error. We will correct this in the revised version.

Line 57ff: "Many studies ...". Sentence seems out of place here, move above as the paragraph above seems to be the introduction to the 27-day variability. Also please give some references to the "many studies".
Response: We will merge this sentence as per reviewer suggestions and include references.

Line 59: Since direct EUV measurements ... and are still not available in the full spectrum...: In recent times the situation of the EUV measurements has considerably improved (thanks to e.g. SDO/EVE, see also http://lasp.colorado.edu/lisird/). Also, while degradation of space instruments is still a challenge, the availability of SSI data in the EUV (either direct measurements, composite datasets or models) has improved, see e.g. Lean et al. http://adsabs.harvard.edu/abs/2003JGRA..108.1059L Haberreiter et al., 2017, composite covers the full spectrum, incl. the EUV Please revisit the statement.
Response: We agree with the reviewer's suggestions. It will be improved in the revised version of the manuscript.

Line 63: .. and indices based on direct EUV measurements (e.g., Unglaub et al., 2011) like the Solar EUV Experiment (SEE) onboard the Thermosphere Ionosphere Mesosphere Energetics and Dynamics (TIMED) satellite (Woods et al., 2000). -> .. and indices developed by Unglaub et al. (2011) based on direct EUV measurements obtained with the Solar EUV Experiment (SEE) onboard the Thermosphere Ionosphere Mesosphere Energetics and Dynamics (TIMED) satellite (Woods et al., 2000).
Response:  We will replace this sentence as suggested.

Line 65: which may be overcome by repeated calibration -> please clarify what is meant here, inflight calibration is repeated calibration. Do the authors mean rocket calibrations of flight spares as done with SDO/EVE?
Response: Yes, repeated calibration of EUV spectrometers.

Line 93: OMNIWeb Plus database: please give a reference and/or link to the database.
Response: We will add the link and reference as suggested.

Line 103: The zonal mean plot additional temporal variations: please explain which those are.
Response: We will rephrase the sentence with clarification.

 Line 105: around the magnetic equator: the variation seems rather symmetric around the equator. The magnetic equator should possibly be indicated in the plot if possible, or would "the equator" be sufficient.
Response: Thanks for the suggestion. We will add the magnetic equator line in the plot.

Line 107: is varies -> varies
Response: We will improve this in the revised text.

Line 109: delete "E.g." at the beginning of the sentence
Response: We will improve this in the revised text.

Line 110: "As all the time series in Figure 1 show a similar overall variation during the 11-year solar cycle, the fundamental behaviour of solar radiation emission is identical at all the wavelengths." A lot of care needs to be taken here. Actually, the fundamental behaviour is not the same for all wavelength, as the plasma heating and atomic processes are different for different wavelength. Specifically, for the radio proxies the processes for the various proxies are

different          (see          Dudok          de          Wit,          et          al.,          2014,
http://adsabs.harvard.edu/abs/2014JSWSC...4A..06D)
Response: We agree with the reviewer's suggestion. We will rephrase the sentence.

Line 120: Note that the T-I system is not only influenced by solar activity but also by changing geomagnetic conditions due to solar wind variations. -> Please revise, suggestion: Note that the T-I system is not only influenced by the solar electromagnetic radiation but also by changing solar energetic particles and geomagnetic conditions due to solar wind variations or Coronal Mass Ejections reaching the Earth. Please also give reference to support this. Effect of particles on the Earth upper atmosphere?
Response: We will replace this sentence as suggested by the reviewer. We will discuss the effect of particles on the upper atmosphere. Actually, the Sun's energetic particles deposition to the Earth's atmosphere is very complicated, and its interaction with the magnetic field plays an important role since the charged particles are guided along magnetic field lines. Thus low latitudes are shielded from much but not all of the incoming charged particles with most of the energetic particles being guided into the Earth's atmosphere in the polar regions. Hence these particles majorly influence the polar regions.

Line 121: Strong solar activity during solar maxima might induce stronger interaction...: Please revise sentence, suggestion: The response to solar forcing is higher during solar maximum... Please also add references. The solar wind, in particular from coronal holes, also occurs during solar minimum conditions. Please take this also into account.
Response: Thanks for the suggestion. We will revise it in the revised version of the manuscript.

Line 135: This allows to determine dominant joint oscillations -> This allows us to determine dominant correlated oscillations (or other word for "joint")
Response: We will replace this sentence as suggested by the reviewer.

Line 137: 16-32 days period region -> 16 to 32-day interval (also elsewhere in the text)
Response: We will improve this throughout the manuscript in the revised version as suggested by the reviewer.

Line 138: the ionospheric variation due to the solar activity is lower -> the ionospheric variation is lower due to solar activity
Response: We will replace this sentence as suggested by the reviewer.

Line 142: The black arrows in Figure 3 indicate the phase relationship between solar proxies and GTEC (also caption of Figure 3): What does the upward orientation of the arrow mean?
Response: We will add the description of arrows in the revised version of the manuscript.

Line 144: the annual and semi-annual period range -> ranges? Could you give the exact interval for those. It is two separate intervals that are meant here? Please clarify.
Response: Yes, these are two separate intervals: the annual (256 to 512-day) and semi-annual (128 to 256 day). We will add it in the revised version of the manuscript.

Line 153: ... semi-annual. The observed periodicities in GTEC are also shown by Hocke (2008). -> ... semi-annual, which is in line with Hocke (2008) (if this is what is meant).
Response: Yes, we will rephrase the sentence as per suggestion.

Line 154: It is interesting to note here that a 44-day periodicity is observed in GTEC and all other solar proxies.: From Figure 4 the 44-day variability seems not significant. It seems that there is random variability in the window up to 1/2 year. Of the same order of magnitude in the time series. Without further analysis it cannot be stated that a 44-day variability is visible in "all other solar proxies". Please revise.
Response: We agree with the reviewer suggestion. We will include 95% confidence line in the updated Figure. 45 days periodicity is observed in F10.7, Mg II and SSN. Hence, we will revise the statement as per suggestions.

Line 156: .. and it's 2nd harmonic 13.5 days, and 4th harmonic 6.7 days...: Please also indicate these harmonics in Figure 17.
Response: Thanks for the suggestion. We will update the figure as suggested.

Line 157: Here similar kind of oscillations..: Do the authors find the same oscillations, i.e. 2nd harmonic 13.5 days, and 4th harmonic 6.7 days. Or are they different for Lyman alpha. If so, please specify.
Response: We will revise the text. Both Mg II and Ly-alpha shows the same periodicity.

Line 157: Ly-$nalpha$ - Ly-$nalpha$ (take out space)
Response: We will improve it in the revised version.

Line 159ff: Note that the wavelet spectra show some periodicity at the half-year time scale, but with variable phase so that they extinguish in the periodogram.:

This sentence needs to be revisited. For which proxies? In Figure 4 only the GTEC and and maybe F30 show a 1/2 year peak. Please be specific.
Response: We will add specific proxies and revise the sentence.

Line 162: Maybe add a subtitle here: "Wavelet Cross-Correlation"
Response: We will add the title as suggested by the reviewer.

Line 164: using ->based on (repetition from line 163)
Response: We will replace this word in the revised version of the manuscript.

Line 167: The delay is mostly positive or zero, which means that TEC is following the solar proxies with delay. -> The delay is mostly positive or zero, which means that TEC is following the solar proxies.
Response: We will replace this sentence in the revised version of the manuscript.

Line 170: ... by about one day: could you please give the exact value here (and everywhere in the text when the lag is given)? E.g. Line 176, 176
Response: As mentioned above, in this analysis we have used all the datasets (GTEC and Solar proxies) at the daily resolution, as solar proxies are available in daily resolution. So, the calculated delay will be in the day(s) only.

Line 184: A stronger correlation -> A strong correlation
Response: We will replace this in the revised version of the manuscript.

Line 186: for the GTEC -> for GTEC
Response: We will improve this in the revised text.

Line 186: with Daubechies 2 ... -> with the Daubechies
Response: We will improve this in the revised text.

Line 194: There is no strong semi-annual cycle visible. -> ... and as expected, no significant semi-annual cycle is visible.
Response: We will improve this in the revised text.

Line 199: inter-annual time scales: these are timescales of one year or larger? Please clarify. Maybe "time scales below (or above) one year"
Response: Yes, we have considered all the variability below and above one year. We will add this in the revised version.

Line 200: 365 days running window -> 365-day running winding
Response: We will improve this in the revised text.

Line 203: All solar proxies show similar behaviour during low activity conditions: While the temporal variation of the CC for Mg II, Ly alpha and He II is largely similar, the SSN (green curve) shows a significantly different behaviour.
Response: We will revise it.

Line 203: apart from a different mean level: Not sure what the author mean as "mean level". It could be stated that SSN generally shows a lower CC than the other proxies.
Response: We apologize for this error. We will rephrase the text as per reviewers suggestions.

Line 208: Are the cross-correlations shown in Fig. 8 a temporal mean over the years 199 to 2017. It would be very interesting to see the temporal variation, e.g. similar to Fig. 7, if possible.
Response: No, this is maximum cross-correlation calculated at 27 days solar rotation period. We will add a description in the revised version of the manuscript. Thanks for the suggestion, we will add the temporal variation plots for GTEC, low latitude, mid-latitude, and high latitude TEC in the appendix.

Line 210ff: "As in Figure 7, the correlation of F10.7 with TEC is weaker than the one of MG-II and TEC.". However, F10.7 is not shown in Fig. 7. Please revise.
Response: We apologize for this error. We will revise it in the revised version.

Line 210ff (disicussion of Fig. 8). Maybe start out with: Generally, the correlation coefficient and the lag for the Global, NH, SH, LL, and ML are very close. Then continue: The maximum correlation is found... The weakest correlation is observed...
Response: We will improve this in the revised text.

Line 211: HL with maximum correlation coefficients -> HL with a maximum correlation coefficient of
Response: We will improve this in the revised text.

Line 215: "response time of about two and one days": As already mentioned above, could the authors give a more precise result for the lag, i.e. derived from the maximum of the curves in Fig. 8. It might be something like 1.8 or 1.9. Please also mark the maximum of each curves in Fig. 8 with a cross or similar.
Response: We will mark the maximum of each curve with a cross.

Line 225: is shown -> is found
Response: We will improve this in the revised text.

Line235: From the above discussion it is clear that during -> In summary, during low solar activity... ; please also add a summary sentence for high solar activity
Response: We will improve this in the revised text.

Line 238: indices shows stronger -> indices show a stronger
Response: We will improve this in the revised text.

Line 239: of about 1-3 days: again, please provide a more precise determinination of the lag.
Response: As mentioned above, in this analysis we have used all the datasets (GTEC and Solar proxies) at the daily resolution, as solar proxies are available in daily resolution. So, the calculated delay will be in the day(s) only.

274: using Empirical Orthogonal Function (EOF) which decomposes -> Empirical Orthogonal Functions (EOFs), which decompose
Response: We will improve this in the revised text.

Line 276: to represents -> to represent
Response: We will improve this in the revised text.

Line 282: How do the authors derive the contributions of the PC1, PC2, ect of about 86%, 11%, ect. Please add this to the text.
The result of EOF2, 11%, is given in brackets in line 282 and 292 again. The first mention could be removed, as the text is then better to read.
Response: We will add the description of PC extraction and rephrase the paragraph as per the reviewer suggestion in the revised version of the manuscript.

Line 299: remove "only", as both semi-annual and annual oscillations are visible.
Response: We will remove this word.

Lien 302: In order to check the relation between solar proxies and geomagnetic parameters (daily Kp, Dst, and Ap indices) with PCs corresponding to EOFs, cross-correlation and delay is calculated and shown in Figure 12. The color-coded value in Fig. 12 is the temporal average of the correlation coefficient? This should be stated in the text and the figure caption. Figure 12 (and Figure 14) are very interesting result of the paper. Therefore, it would be very interesting to also see the temporal variability of the correlation (also similar to Fig. 7 for the solar proxies). Could the authors provide this for completeness (at least for a few cases), possibly in the appendix?
Response: We will add temporal variability plots for all four PCs for Figure 12 in the appendix.

Line 315: is capturing -> captures

Response: We will improve this in the revised text.

Figures: Figures in general: a number of panels are rather on the small side. It is recommended to fill the full text width in order to make the figures better readable Figure 1, 3, 8, 10, 11 : enlarge panels to 0.5 textwidth
Response: Thanks for the suggestions. We will improve all the Figures as per the reviewer suggestions.

Figure 11: Panels need to be enlarged, as it is difficult to see the details in the printed version of the manuscript. The panels should be of the size of the new Fig. 10, i.e. only two panels next to each other. Could Figure 11 be split, say PC1 and PC2 in one figure, and PC3/PC4 in another?
Response: Thanks for the suggestions. We will split Figure 11 into two Figures as suggested by the reviewer.

Caption Figure 8: please add: Temporal mean (cross correlation .... during the years 1999 to 2017) for different lags.
Response: We will improve the caption.

Caption Figure 9: The background colors show the correlation coefficient -> The background colors give the temporal mean of the correlation coefficient (or maximum?), please clarify which cc is shown throughout the text.
Response: Thanks for the suggestions. We will improve the Figures and captions as per the reviewer suggestions.

Throughout the text: He-II-> He II (no hyphen) MG-II-> Mg II (no hyphen) CaK -> Ca II K 16-32 days period -> 16 to 32-day period (day always without s) thoughout the text and figure captions The authors often show a correlation coefficient (Figure 8, Figs 12, 14, ect). Please specify if it is a temporal mean, spatial mean, maximum value corresponding to a lag value? Use Figure xx (at the beginning of the sentence, as in Line 106) and its abbreviation Fig. xx (in sentence, as e.g. in line 105) consistently throughout the text. Generally, the second part of the paper, starting with lines 208 reads more fluent. Could the (co-)authors go over the manuscript for language and typos.
Response: Thanks for the suggestions. We will improve the text as per the reviewer suggestions.

---

## Author Comment (AC4) · 7 Aug 2019

Answer to Reviewer #1:

We are thankful for the reviewer's comments and suggestions which help us to improve the quality of the manuscript. We have revised the paper according to the suggestions and comments.

We discussed most of the comments of reviewer#1 in the first response. Here we summarise the important points which we have included in the revised version. The blue color refers to the comments of reviewer#1. Few common points from both the reviewers marked with red or blue color.

General comments about the manuscript

In Figure 1b and Figure 4 parameters do not separated easily, please use different colors as much as possible for each parameter. In the current version especially red and pink colors are mixing.
Response: The Figures have been modified in the revised version. Page 19-20

All abbreviations should be described clearly in the first place that they appear in the manuscript. In the current version of the manuscript some of them are not given with full name. Also, for the daily sunspot area the abbreviation is given as DSA. Please replace it as daily SSA
Response: We added the abbreviation and replace DSA with SSA in the revised version. Page 1, Line 8-10

In Figure 4 the significance levels of obtained periodicities are not given. I suggest that authors should add at least 95 % confidence level line to each periodogram.
Response: We added this in the revised version. Page 20

Please add some information about the appendix figures inside the manuscript.
Response: We added the description of appendix figures in the revised version.

Page 1 line 21, authors mentioned that "Wavelet variance estimation suggests that GTEC variance is highest for the seasonal timescale followed by the 16-32 days period, similar to the F10.7 index highest variance for the 16-32 days period." Please replace as "Wavelet variance estimation suggests that GTEC variance is highest for the seasonal timescale followed by the 16-32 days period, similar to the F10.7 index.
Response: Thanks, this has been replaced in the revised version. Page 1 line 20-22

Line 25 "DSA" – "Daily SSA"
Response: Done. Line 25

Line 34 "(e.g. Schmölter et al., 2018)", please add a few more reference.
Response: We added more references in the revised version. Line 37

Page 2 line 55, ": : :at different time scales." – "at different time scales such as (: : :)." Please clarify
Response: We has been corrected this in the revised manuscript. Line 62

Page 4 line 136 ": : :GTEC with four selected solar proxies: : :" please give these solar proxies inside a parenthesis.
Response: We added these proxies as suggested. Page 5, line 154

In page 5 line 157, authors mentioned that they used 7 days smoothed data and they mentioned 6.7 days periodicity. From 7 days smoothed data it is not possible to get 6.7 days periodicity. This part should be removed.
Response: We removed this part of the sentence in the revised version.

Authors mentioned 128 – 256 days periodicity from GTEC and solar parameters. Source of this periodicity should be given more clearly (see Lou et al. 2003, Kilcik et al, 2018). For the 45 days periodicity, it is also one of the fundamental periodicity of solar activity and it detected in many solar activity indices (Lou et al. 2003, Chowdhury et al. 2015, Kilcik et al, 2018).
Please explain this periodicity a bit more detail.
(Lou, Y.Q., Wang, Y.M., Fan, Z., Wang, J.X., Wang, S.: 2003, Mon. Not. Roy. Astron. Soc. 345, 809.
Chowdhury, P., Choudhary, D.P., Gosain, S., Moon, Y.J.: 2015, Astrophys. Space Sci. 356, 7.
Kilcik, A., Yurchyshyn, V., Donmez, B., Obridko, V.N., Ozguc, A., Rozelot, J.P.: 2018, Solar Phys. 293, 63.)
Response: Thank you for the suggestion. We added a description of the sources of periodicities in the revised manuscript. Page 5-6, line 175-181, 187-192

In page 6 line 179, authors mentioned that ": : :solar rotation period of 27 days is only a mean value and different solar regions rotate with a different velocity which can be up to 35 days." Please replace this sentence as ": : :the 27 days periodicity is only a mean value of solar differential rotation. It also strongly depends on the life time and proper motion of observed active regions."
Response: Done. Page 6-7, line 211-212

Page 6 line 204, "The correlation coefficient is also decreasing during high solar activity years such as 2002 and 2014 but increases during the recovery phase of solar activity." This sentence is not correct, it should be clarified.

Response: Thanks for your observation, and we improved the description in the revised version and added modified figures for short and intra-annual timescales. Page 7-8, line 231-254

Page 8 line 246, authors mention that "The F1.8 and DSA cannot adequately represent the solar activity at the solar rotation (16-32 days) time scale." SSA is one of the best solar indicator in solar physics literature, so please clarify this sentence with more detail.

Response: We added more explanation in the revised version. Page 8, line 254-259, Page 9, line 257

In line 264, ": : :several other physical processes." Please clarify these processes

Response: We added these processes in the revised version. Page 9, line 314-316

In general, please use wavelet scalogram instead of wavelet transforms for wavelet plots. Also in the wavelet plots, what is the meaning of negative power it should be explained clearly or wavelet scalograms should be modified.

Response: We added the description of negative power in the revised version of the manuscript. Page 10, Line 344-345

I think current version of the manuscript is not appropriate for the publication in the journal. It needs some corrections.

Response: Thank you for reviewing our manuscript. We have revised the manuscript according to the comments and suggestions.

[revised manuscript text omitted]

---

## Author Comment (AC5) · 7 Aug 2019

Answer to Reviewer:

We are thankful to the reviewer's comments and suggestions which help us to improve the quality of the manuscript. We have revised the paper according to the suggestions and comments.

We discussed most of the comments of the reviewer in the first response. Here we summarise the important points which we have included in the revised version. The green color refers to the comments of the reviewer.

1. You are using global TEC from GIM maps. However, there is a jump in GIM TEC in 2001, the values before being lower than values after (see Emmert et al., JGR-SP, 2017, doi:10.1002/ 2016JA023680). It has probably no effect on your result but for your future work.
Response: We included this in the revised version. Line 118-119

2. The 27-day variation in the lower ionosphere (D-region) is often predominantly caused by dynamical forcing (PW), not by direct solar forcing, particularly in winter under low solar activity (Pancheva et al., JATP, 1991, https://doi.org/10.1016/0021-9169(91)90064-E).
Response: We included this in the revised version. Line 172-175

[revised manuscript text omitted]

---

## Author Comment (AC6) · 7 Aug 2019

Answer to Reviewer #2:

We are thankful for the reviewer's comments and suggestions which help us to improve the quality of the manuscript. We have revised the paper according to the suggestions and comments.

We discussed most of the comments of reviewer#2 in the first response. Here we summarise the important points which we have included in the revised version. The red color refers to the comments of reviewer#2. Few common points from both the reviewers marked with red or blue color.

Major Comments: ================

The authors present new and also very interesting results. However, additional clarifications are still to recommend the paper for publication. The result of the lag is presented in the text mainly as lag of one or two days. As the result of the lag is also an important result it needs to be presented more precisely, i.e. with at least one (or better two) digits after the comma, i.e. 1.8 (e.g. from Figure 8), corresponding to maximum of the cc-curve?

Response: The authors are thankful to the reviewer for appreciating our findings. In this analysis, we have used all the datasets (GTEC and Solar proxies) at a daily resolution, as our used solar proxies are available in daily resolution. So, the calculated delay is in the day(s) only. We marked the maximum of cc curves in Fig. 8, where the delay is 1 or 2 days. Page 23

In the abstract (Line 26ff) the authors state that "Empirical orthogonal function (EOF) analysis of the TEC data shows that the first EOF components capture more than 86% of the variance, and the first three EOF components explain 99% of the total variance." The authors should specify who the contributions (86%, ect) are determined. The authors state that the first EOF is the solar component. Could the authors elaborate on the other EOFs under consideration (in particular the 2nd and 3rd). This is partly done in lines 192ff. Could the dynamics of the Earth's atmosphere also play a role?

Response: We added a brief description of EOFs analysis. Line 28-30, Line 329-336

The link between the EOFs and PCs is not clear. In Figure 11 the authors plot EOF1 to EOF4. In Figure 12 and 14 the authors show the CC of the PCs with proxies, and in Lines 302 the authors state "In order to check the relation between solar proxies and geomagnetic parameters (daily Kp, Dst, and Ap indices) with

PCs corresponding to EOFs, cross-correlation and delay is calculated and shown in Figure 12." Could the authors elaborate better how the EOFs and PCs are derived and what is the time series for the CC in Figs 12 and 14.

Response: We added a paragraph describing the link between EOFs and PCs in section 3.6. Line 329-336

In the introduction, further references to previous work should be mentioned e.g.: http://adsabs.harvard.edu/abs/2016JGRA..12110367L and others.

Response: We added more references in the revised version. Line 54-57

Minor comments: ===============

Line 2: please clarify or rephrase "spatial dynamic of solar activity". The solar proxies under consideration do contain any spatial information, possibly the authors mean "the spatial response of the ionosphere to solar activity"?

Response: We rephrased this sentence as suggested. Line 8-9
.
Line 10: GNSS, explain acronym when first mentioned

Response: We added the acronym. Line 10

Line 35ff: "These studies have shown, that the response of the ionosphere to solar EUV radiation variations takes 1-2 days for solar radiation changes within 27 days solar rotation period". This sentence is not clear, could the authors please rewrite it.

Response: We corrected this in the revised manuscript: "These studies have shown that the response of the ionosphere to solar EUV radiation variations gets delayed by 1-2 days at 27 days solar rotation period". Page 2, Line 39-40

Line 13: A 16-32 days period -> A 16 to 32-day period (day, without s)

Response: Done. Line 13

Line 15: LSP analysis -> The LSP analysis

Response: Done. Line 16

Line 18: "The wavelet variance estimation method is used to find the variance in the maximum of the solar cycles (SC) 23 (2000-2002) and 24 (2012-2014), for GTEC and F10.7 index, respectively. " Suggested rephrasing, as the sentence does not read very well. -> The wavelet variance estimation method is used to find the variance of GTEC and F10.7 over the maxima of the solar cycles SC 23 and SC 24. The selected time frame that covers the solar maxima are .... and ....

Response: Done. Line 19-20

Line 20: GTEC variance -> the GTEC variance
Response: Done. Line 20

Line 20: seasonal timescale: which one is considered as the seasonal time scale? 32-64-day period? please specify or rather give the name of the wavelet window. Generally, the wavelet intervals could be numbered so that the interval does not need to be repeated in the text again.
Response: We added this in the revised version of the manuscript. Line 20

Line 22: to represent the solar activity -> to represent solar activity
Response: Done.

Line 23: may be placed at the second ... -> may be placed second ...
Response: Done. Line 23-24

Line 24: but there are some differences between solar maximum and minimum: could the authors be more specific.
Response: We rephrased the sentence in the revised version of the manuscript. Line 24-25

Line 25: The F1.8 and DSA ... -> The indices F1.8 and DSA ...
Response: Done.

Line 26: Empirical orthogonal function (EOF) analysis -> The Empirical orthogonal function (EOF)
Response: Done.

Line 27: "EOF analysis suggests that the first component is associated with the solar flux." This result is expected, but also very nice to be an outcome of the EOF analysis. Could the authors also indicate what the status of the knowledge/hypothesis about the nature of the subsequent 2-3 EOF components are (dynamics, ect).
Response: We have included the nature of EOF2/3 in the abstract. Line 28-30

Line 33: reference Chen et al., 2012: Please add more references.
Response: We added more references in the revised version. Line 35

Line 43: Investigate ... mechanism -> investigate the ... mechanism
Response: Done. Line 46

Line 48: "The T-I system is also influenced by different external forces": the solar forcing should also be considered as "external forcing". Therefore, aren't all forcings "external"?

Response: We corrected it by rephrasing the sentence in the revised manuscript. Line 51-53

Line 49: "In the case of solar events, the forcing from above might even result in strong disturbances affecting the ionospheric delay." -> This sentence needs to be revised. Suggestion: In addition to the solar EUV forcing, the solar wind as well as solar eruptions might also result in ... Could the authors give references that address this work?

Response: We rephrased the sentence as suggested. Line 51-53

Line 50: "As a result, the ionospheric plasma behaviour is varying during different solar activity conditions." It is not clear what is meant here. Please revise this sentence.

Response: We have rephrased the sentence. Line 53-54

Line 51-58 (full paragraph): The authors mention the 27-day solar rotation period and its effects on the TEC. What is missing in Hocke (2008). Why are further investigations needed?

Response: We corrected it in the revised version. Line 58-63

Line 57ff: "Many studies ...". Sentence seems out of place here, move above as the paragraph above seems to be the introduction to the 27-day variability. Also please give some references to the "many studies".

Response: We merged this sentence as per reviewer suggestions and include references. Line 39-41

Line 59: Since direct EUV measurements ... and are still not available in the full spectrum...: In recent times the situation of the EUV measurements has considerably improved (thanks to e.g. SDO/EVE, see also http://lasp.colorado.edu/lisird/). Also, while degradation of space instruments is still a challenge, the availability of SSI data in the EUV (either direct measurements, composite datasets or models) has improved, see e.g. Lean et al. http://adsabs.harvard.edu/abs/2003JGRA..108.1059L Haberreiter et al., 2017, composite covers the full spectrum, incl. the EUV Please revisit the statement.

Response: We rewrote the paragraph in the revised version of the manuscript. Line 65-72

Line 63: .. and indices based on direct EUV measurements (e.g., Unglaub et al., 2011) like the Solar EUV Experiment (SEE) onboard the Thermosphere Ionosphere Mesosphere Energetics and Dynamics (TIMED) satellite (Woods et

al., 2000). -> .. and indices developed by Unglaub et al. (2011) based on direct EUV measurements obtained with the Solar EUV Experiment (SEE) onboard the Thermosphere Ionosphere Mesosphere Energetics and Dynamics (TIMED) satellite (Woods et al., 2000).
Response: Done. Line 70-72

Line 93: OMNIWeb Plus database: please give a reference and/or link to the database.
Response: We added the link and reference as suggested. Line 96

Line 103: The zonal mean plot additional temporal variations: please explain which those are.
Response: We rephrased the sentence. Line 109-110

 Line 105: around the magnetic equator: the variation seems rather symmetric around the equator. The magnetic equator should possibly be indicated in the plot if possible, or would "the equator" be sufficient.
Response: We added the magnetic equator line in the plot. Page 20

Line 107: is varies -> varies
Response: Done.

Line 109: delete "E.g." at the beginning of the sentence
Response: Done.

Line 110: "As all the time series in Figure 1 show a similar overall variation during the 11-year solar cycle, the fundamental behaviour of solar radiation emission is identical at all the wavelengths." A lot of care needs to be taken here. Actually, the fundamental behaviour is not the same for all wavelength, as the plasma heating and atomic processes are different for different wavelength. Specifically, for the radio proxies the processes for the various proxies are different (see Dudok de Wit, et al., 2014, http://adsabs.harvard.edu/abs/2014JSWSC...4A..06D)
Response: We have corrected it in the revised version. Line 122-126

Line 120: Note that the T-I system is not only influenced by solar activity but also by changing geomagnetic conditions due to solar wind variations. -> Please revise, suggestion: Note that the T-I system is not only influenced by the solar electromagnetic radiation but also by changing solar energetic particles and geomagnetic conditions due to solar wind variations or Coronal Mass Ejections reaching the Earth. Please also give reference to support this. Effect of particles on the Earth upper atmosphere?

Response: We rewrote the paragraph, including the suggestions and added references. Line 134-142

Line 121: Strong solar activity during solar maxima might induce stronger interaction...: Please revise sentence, suggestion: The response to solar forcing is higher during solar maximum... Please also add references. The solar wind, in particular from coronal holes, also occurs during solar minimum conditions. Please take this also into account.
Response: We rewrote the paragraph, including the suggestions and added references. Line 134-142.

Line 135: This allows to determine dominant joint oscillations -> This allows us to determine dominant correlated oscillations (or other word for "joint")
Response: Done. Line 152

Line 137: 16-32 days period region -> 16 to 32-day interval (also elsewhere in the text)
Response: Done. Line 154 and throughout the text

Line 138: the ionospheric variation due to the solar activity is lower -> the ionospheric variation is lower due to solar activity
Response: Done. Line 156

Line 142: The black arrows in Figure 3 indicate the phase relationship between solar proxies and GTEC (also caption of Figure 3): What does the upward orientation of the arrow mean?
Response: We added the description of arrows in the revised version of the manuscript. line 160-162

Line 144: the annual and semi-annual period range -> ranges? Could you give the exact interval for those. It is two separate intervals that are meant here? Please clarify.
Response: We added it in the revised version of the manuscript. Line 163

Line 153: ... semi-annual. The observed periodicities in GTEC are also shown by Hocke (2008). -> ... semi-annual, which is in line with Hocke (2008) (if this is what is meant).
Response: Done. Line 171

Line 154: It is interesting to note here that a 44-day periodicity is observed in GTEC and all other solar proxies.: From Figure 4 the 44-day variability seems not significant. It seems that there is random variability in the window up to 1/2 year. Of the same order of magnitude in the time series. Without further analysis

it cannot be stated that a 44-day variability is visible in "all other solar proxies". Please revise.

Response: We have added the revised paragraph for 45-day periodicity in the revised version. Line no.: 175-181

Line 156: .. and it's 2nd harmonic 13.5 days, and 4th harmonic 6.7 days...: Please also indicate these harmonics in Figure 17.

Response: We updated the figure as suggested. Page 21

Line 157: Here similar kind of oscillations..: Do the authors find the same oscillations, i.e. 2nd harmonic 13.5 days, and 4th harmonic 6.7 days. Or are they different for Lyman alpha. If so, please specify.

Response: We found the same oscillations.

Line 159ff: Note that the wavelet spectra show some periodicity at the half-year time scale, but with variable phase so that they extinguish in the periodogram.: This sentence needs to be revisited. For which proxies? In Figure 4 only the GTEC and and maybe F30 show a 1/2 year peak. Please be specific.

Response: We added specific proxies. Line 186

Line 162: Maybe add a subtitle here: "Wavelet Cross-Correlation"

Response: Done. Line 193

Line 164: using ->based on (repetition from line 163)

Response: Done. Line 196

Line 167: The delay is mostly positive or zero, which means that TEC is following the solar proxies with delay. -> The delay is mostly positive or zero, which means that TEC is following the solar proxies.

Response: Done. Line no.199-200

Line 184: A stronger correlation -> A strong correlation

Response: Done.

Line 186: for the GTEC -> for GTEC

Response: Done.

Line 186: with Daubechies 2 ... -> with the Daubechies

Response: Done.

Line 194: There is no strong semi-annual cycle visible. -> ... and as expected, no significant semi-annual cycle is visible.

Response: Done. Line 226-227

Line 199: inter-annual time scales: these are timescales of one year or larger? Please clarify. Maybe "time scales below (or above) one year"
Response: We rewrote the paragraph including both the reviewer's suggestions. Line 230-258.

Line 200: 365 days running window -> 365-day running winding
Response: Done. Line 237

Line 203: All solar proxies show similar behaviour during low activity conditions: While the temporal variation of the CC for Mg II, Ly alpha and He II is largely similar, the SSN (green curve) shows a significantly different behaviour.
Response: Done. We rewrote the paragraph, including both the reviewer's suggestions. Line 230-258.

Line 203: apart from a different mean level: Not sure what the author mean as "mean level". It could be stated that SSN generally shows a lower CC than the other proxies.
Response: Done. Line 248-250

Line 208: Are the cross-correlations shown in Fig. 8 a temporal mean over the years 199 to 2017. It would be very interesting to see the temporal variation, e.g. similar to Fig. 7, if possible.
Response: We added the temporal variation plots for GTEC, low latitude, mid-latitude, and high latitude TEC in the appendix. Page 28

Line 210ff: "As in Figure 7, the correlation of F10.7 with TEC is weaker than the one of MG-II and TEC.". However, F10.7 is not shown in Fig. 7. Please revise.
Response: Done.

Line 210ff (disicussion of Fig. 8). Maybe start out with: Generally, the correlation coefficient and the lag for the Global, NH, SH, LL, and ML are very close. Then continue: The maximum correlation is found... The weakest correlation is observed...
Response: Done. Line 261-265

Line 211: HL with maximum correlation coefficients -> HL with a maximum correlation coefficient of
Response: Done. Line 264

Line 225: is shown -> is found
Response: Done. Line 275

Line235: From the above discussion it is clear that during -> In summary, during low solar activity... ; please also add a summary sentence for high solar activity
Response: Done. Line 288-289

Line 238: indices shows stronger -> indices show a stronger
Response: Done.

274: using Empirical Orthogonal Function (EOF) which decomposes -> Empirical Orthogonal Functions (EOFs), which decompose
Response: Done. Line 327-328

Line 276: to represents -> to represent
Response: Done.

Line 282: How do the authors derive the contributions of the PC1, PC2, ect of about 86%, 11%, ect. Please add this to the text.
The result of EOF2, 11%, is given in brackets in line 282 and 292 again. The first mention could be removed, as the text is then better to read.
Response: We added the description of PC extraction and rephrase the paragraph as per the reviewer suggestion in the revised version of the manuscript. Line 329-336

Line 299: remove "only", as both semi-annual and annual oscillations are visible.
Response: Done.

Lien 302: In order to check the relation between solar proxies and geomagnetic parameters (daily Kp, Dst, and Ap indices) with PCs corresponding to EOFs, cross-correlation and delay is calculated and shown in Figure 12. The color-coded value in Fig. 12 is the temporal average of the correlation coefficient? This should be stated in the text and the figure caption. Figure 12 (and Figure 14) are very interesting result of the paper. Therefore, it would be very interesting to also see the temporal variability of the correlation (also similar to Fig. 7 for the solar proxies). Could the authors provide this for completeness (at least for a few cases), possibly in the appendix?
Response: We added temporal variability plots for all four PCs for Figure 12 in the appendix. Page 29

Line 315: is capturing -> captures
Response: Done. Line 349

Figures: Figures in general: a number of panels are rather on the small side. It is recommended to fill the full text width in order to make the figures better readable

Figure 1, 3, 8, 10, 11 : enlarge panels to 0.5 textwidth

Response: Thanks for the suggestions. We improved all the Figures as per the reviewer suggestions.

Caption Figure 8: please add: Temporal mean (cross correlation .... during the years 1999 to 2017) for different lags.

Response: Done. Line 652-654

Caption Figure 9: The background colors show the correlation coefficient -> The background colors give the temporal mean of the correlation coefficient (or maximum?), please clarify which cc is shown throughout the text.

Response: Done. Line 661

Throughout the text: He-II-> He II (no hyphen) MG-II-> Mg II (no hyphen) CaK -> Ca II K 16-32 days period -> 16 to 32-day period (day always without s) thoughout the text and figure captions The authors often show a correlation coefficient (Figure 8, Figs 12, 14, ect). Please specify if it is a temporal mean, spatial mean, maximum value corresponding to a lag value? Use Figure xx (at the beginning of the sentence, as in Line 106) and its abbreviation Fig. xx (in sentence, as e.g. in line 105) consistently throughout the text. Generally, the second part of the paper, starting with lines 208 reads more fluent. Could the (co-)authors go over the manuscript for language and typos.

Response: Thanks for the suggestions. We improved the text in the revised version.

[revised manuscript text omitted]

---

## Author Response (AR2)

We are thankful for the reviewer's comments and suggestions, which help us to improve the quality of the manuscript. We have revised the paper according to the suggestions and comments.

Answer to Reviewer #1:

A few small corrections
1. Line 110 to clarify please mentioned that "solar activity was very low during SC24 compared to SC23"
Response: We added this in the revised version. Page 4

2. For the correlation instead of "poor" please use "weak" along the text
Response: Thanks, this has been replaced in the revised version.

3. line 351 "capture almost 97 %.. " should be "...98
Response: This has been corrected in the revised version. Page 11

Typo's correction:
In figure 8. "NH is replaced by SH.

Anonymous Referee #3

The overall paper is very interesting and acceptable for publication, although it enters in so many details that you sometimes miss their point. Statistics in full detail, as in this case, use to have this effect, in my opinion. I found particularly interesting the analysis of the correlation between TEC and MgII analyzed in terms of location (shown in Figure 10 b), where it is clearly seen that going to higher latitudes the correlation is lower.

I found only one thing that bothers me a little in Figure 8. If Global TEC values are the average between SH and NH, and also the average between LL, ML and HL, I would expect that the correlation coefficient of Global TEC with F10.7 and with Mg II) fall in the middle, or somewhere between the correlation between NH and NH TEC with F10 and Mg II, and also in between the correlation coefficient between the other TECs.

Response: Thank you for this hint. As the global TEC values are the weighted averages between SH and NH or LL, ML, HL, the global TEC values will be lower than for LL and greater than for ML, HL. Since the majority of the ionospheric electron content is found at LL, it is expected that the global TEC and its variability is mainly determined by LL TEC, and therefore the LL and global correlation coefficients are similar. We add a sentence at page 8 "Generally, GTEC variability is mainly determined by the LL electron content so that it is expected that the correlation coefficients for GTEC and LL are similar."

The correlation coefficients of GTEC and solar activity are not a linear function of the subtotal correlation coefficients. So, it is possible that deviations from a perfect match at different latitudes may either cancel or add to each other, so that from the correlations at different latitudes the global correlation cannot directly be concluded.

[revised manuscript text omitted]